



# On the predictability of turbulent fluxes from land: PLUMBER2 MIP experimental description and preliminary results

Gab Abramowitz[1,2], Anna Ukkola[1,2], Sanaa Hobeichi[1,2], Jon Cranko Page[1,2], Mathew Lipson[3], Martin G. De Kauwe[4], Samuel Green[1,2], Claire Brenner[5], Jonathan Frame[5], Grey Nearing[6], Martyn Clark[7], Martin Best[8], Peter Anthoni[9], Gabriele Arduini[10], Souhail Boussetta[10], Silvia Caldararu[11,12], Kyeungwoo Cho[13], Matthias Cuntz[14], David Fairbairn[10], Craig R. Ferguson[15], Hyungjun Kim[16], Yeonjoo Kim[13], Jürgen Knauer[17,18], David Lawrence[19], Xiangzhong Luo[20], Sergey Malyshev[21], Tomoko Nitta[22], Jerome Ogee[14], Keith Oleson[19], Catherine Ottlé[23], Phillipe Peylin[23], Patricia de Rosnay[10], Heather Rumbold[8], Bob Su[24], Nicolas Vuichard[23], Anthony P. Walker[25], Xiaoni Wang-Faivre[23], Yunfei Wang[24], Yijian Zeng[24]

[1] CLEX, UNSW Sydney, Australia
[2] CCRC, UNSW Sydney, Australia
[3] Bureau of Meteorology, Australia
[4] School of Biological Sciences, University of Bristol, Bristol, BS8 1TQ, UK
[5] University of Alabama, USA
[6] Google, USA
[7] University of Calgary, Canada
[8] UKMO, UK
[9] Karlsruhe Institute of Technology, Institute of Meteorology and Climate Research/Atmospheric Environmental Research, 82467 Garmisch-Partenkirchen, Germany
[10] European Centre for Medium-Range Weather Forecasts (ECMWF), UK
[11] Max Planck Institute for Biogeochemistry, Jena, Germany
[12] Discipline of Botany, School of Natural Sciences, trinity College Dublin, Dublin, Ireland
[13] Yonsei University, Seoul, Korea
[14] INRAE, France
[15] Atmospheric Sciences Research Center, University at Albany, State University of New York, Albany, NY, USA
[16] HydroKlima Lab, KAIST, Daejeon, Korea
[17] CSIRO Environment, Australia
[18] Western Sydney University, Australia
[19] NCAR, USA
[20] National University of Singapore, Singapore
[21] GFDL, USA
[22] The University of Tokyo, Japan
[23] LSCE, France
[24] University of Twente, Netherlands
[25] Oak Ridge National Laboratory, USA

*Correspondence to*: Gab Abramowitz (gabriel@unsw.edu.au)

**Abstract.** Accurate representation of the turbulent exchange of carbon, water, and heat between the land surface and the atmosphere is critical for modelling global energy, water, and carbon cycles, both in future climate projections and weather forecasts. We describe a Model Intercomparison Project (MIP) that compares the surface turbulent heat flux predictions of



around 20 different land models provided with in-situ meteorological forcing, evaluated with measured surface fluxes using quality-controlled data from 170 eddy-covariance based flux tower sites.

Several out-of-sample empirical model predictions of site fluxes are used as benchmarks to quantify the degree to which land model performance could improve across a broad range of metrics. The performance discrepancy between empirical and mechanistic model predictions also provides a potential pathway to understand sources of model error. Sites with unusual behaviour, complicated processes, poor data quality or uncommon flux magnitude will be more difficult to predict for both mechanistic and empirical models.


Results suggest that latent heat flux and net ecosystem exchange of $CO_2$ are better predicted by land models than sensible heat flux, which at least conceptually would appear to have fewer physical processes controlling it. Land models that are implemented in Earth System Models also appear to perform notably better than stand-alone ecosystem (including demographic) models, at least in terms of the fluxes examined here.


Flux tower data quality is also explored as an uncertainty source, with the difference between energy-balance corrected versus raw fluxes examined, as well as filtering for low wind speed periods. Land model performance does not appear to improve with energy-balance corrected data, and indeed some results raised questions about whether the correction process itself was appropriate. In both cases results were broadly consistent, with simple out-of-sample empirical models, including linear

regression, comfortably outperforming mechanistic land models. The PLUMBER2 approach, and its openly available data, enable precise isolation of the locations and conditions in which model developers can *know* that a given land model can improve, allowing information pathways and discrete parametrisations in models to be identified and targeted for model development.

## 1 Introduction

Land models (LMs) simulate terrestrial water, energy and biogeochemical cycles. They simulate the exchange of heat and moisture with the atmosphere inside weather forecast models, soil moisture and streamflow in hydrological and agricultural applications, ecological dynamics and carbon exchange in ecosystem modelling, and most of these processes combined inside climate models. The fidelity of LM simulations is therefore consequential economically, socially and environmentally.

Given these LM applications, we have reason to ask 'what makes a good land model?' Too often, our answer has been something akin to "the best performing model", or anything "better than the previous version" of a model. These are unsatisfactory answers for a number of reasons, not least that both discount the possibility that all the models we have are poor or unfit for a given application. The key question is "How can we do better?"



Model benchmarking, as opposed to model evaluation, requires setting expectations of model performance *a priori*, before we know how well our model performs. There are obviously many different ways we could choose to do this. We may require qualitative performance improvement relative to a simpler model, for example. Or, if a model's application is well known, and threshold levels of performance in particular metrics can decide what is sufficient, then the answer is relatively clear - a model is good enough when these thresholds are met. Yet this approach still only tells us about the suitability of the model for a given

application, rather than understand whether we have a good model *per se*. What a 'good model' means, without a given application seems like an entirely subjective question, but we suggest this should relate to the fidelity of its representation, given its complexity.

Perhaps the most universal approach to answering this broader question is to understand whether a model does a good job of

utilising the information that is available to it for prediction (in its input time series, parameters and initial states). A 'perfect model' with all required driving variables would, for example, tell us exactly which aspects of observed site behaviour were predictable, and which were not - it would define predictability. But we might also imagine a 'perfect model' that is only provided with a subset of the required driving variables, and expect that it performs with less fidelity. It would still nevertheless define predictability, but conditional on only a subset of predictors being available. We would, for example, expect that a LM

would provide a more sophisticated prediction of evapotranspiration than a response to incoming shortwave radiation alone, since it contains information about soil moisture availability, soil temperature, vegetation and evaporative demand. This is the approach to defining utilisation of information by land models that has been most commonly used - comparing land models to a range of empirical models of increasing complexity, using in-sample empirical approaches to compare to land models calibrated on evaluation data (e.g. eddy covariance fluxes) and out-of-sample empirical approaches to compare to land models

without calibration on evaluation data (e.g. Abramowitz, 2005; Best et al. 2015; Whitley et al., 2017). In our case, the best possible utilisation of information towards flux prediction at a flux tower site defines site predictability.

The Protocol for the Analysis of Land Surface Models (PALS) Land Surface Model Benchmarking Evaluation Project (PLUMBER; Best et al. 2015; Haughton et al., 2016) explored some of these questions in the form of a model comparison

experiment, using 20 flux tower sites and simulations at these sites from nine land surface models (LSMs), driven by half hourly meteorology measured at the sites. Here we detail an enhanced reincarnation of that experiment - PLUMBER2 - that offers a more comprehensive approach than the original experiment in a number of ways.

Firstly, and perhaps most trivially, many of the assumptions and sample size issues in the first experiment are addressed in

PLUMBER2. Data is taken from 170 flux tower sites, covering a much wider range of biomes with many more site-years of data, including several sites with much longer records (>10 years). Next, the data quality control process is considerably more thorough, documented and peer reviewed (see Ukkola et al., 2022). A much broader range of process-based land models (LMs;



the term we'll use to refer to all mechanistic, as opposed to empirical models) have participated, including LSMs (components of weather or climate models), ecosystem models (LMs focused on carbon dynamics) and hydrological models (LMs focused
on the hydrological cycle). Finally, a couple of key data quality concerns are directly explored. Analyses and empirical approaches utilise both energy-balance corrected and raw observed fluxes, with both being compared. Next, the sensitivity of results to low turbulence periods (typically night-time) in flux data are also tested. The Methodology section explains how each of these points are addressed in more detail.

Perhaps most importantly, at least in terms of understanding site predictability and the capacity for LM improvement, PLUMBER2 also includes a much more sophisticated hierarchy of empirical modelling approaches, from simple regression through to different machine learning techniques. These approaches allow us to try to quantify site predictability, by showing how much of a flux's variability at a site can be predicted by empirical models (given the same input predictors as LMs) out-of-sample - that have not been exposed to data from that site. The discrepancy in performance between these approaches and
a LM can also help us understand the potential for LM improvement and, since the empirical models provide time series predictions akin to LM predictions, additionally identify the specific circumstances where the gap between LM predictions and predictability is greatest.

This is also why, as in the original PLUMBER experiment, LMs were provided with a limited amount of site information and
were not allowed to calibrate to testing site fluxes. The restriction is critical for two main reasons. First, we want to understand LMs' fidelity in a global simulation, where this kind of calibration to local conditions is simply not possible, but key constraints like reference (measurement) height and vegetation type are (ideally) appropriately prescribed. Next, key to understanding whether we have a 'good model' is the generalisability of the model - the insight it provides about the system, and its applicability to systems other than those used to develop it. If a LM *requires* calibration to testing data to be useful, one could
argue we are testing the fitting ability of the LM rather than the insight it provides, so that there may be little to distinguish it from machine learning and other empirical approaches that we already know will perform better (Abramowitz 2012; Beaudry and Renner, 2012; Best et al., 2015; Nearing et al., 2018). Out-of-sample testing for any model, even if only partly empirical, is key to understanding its predictive ability (see Abramowitz et al., 2019), especially when it needs to be applied globally.

We also note that different modelling groups arrive at the default vegetation parameter values used for each vegetation type in different ways. While calibration to flux tower data is often part of that process, we clearly do not suggest that LMs and the empirical models used here are calibrated in the same way. It remains unclear what a strictly controlled comparison like that would deliver, if it were possible. It is also not clear how much benefit in-sample testing site calibration would deliver across the LM cohort shown here, relative to in-sample empirical models, although Abramowitz et al (2008) give us some indication.



The concept of generalisability applies as much to temporal change as to spatial change. Being able to predict systematic changes, like future system behaviour and trends in a changing climate, is clearly one key motivation for building LMs, and so any application to this kind of problem also has the same requirement for out-of-sample testing, especially in the absence of fidelity in present conditions.


Since many of the LMs in this experiment are most often run at a global scale on coarse grids (typically 0.5-1°), it also seems reasonable to question whether an evaluation experiment using site-based data provides a fair test of models' ability. There are obvious differences in the nature of fluxes at the site scale of the flux tower data (typically < 1km²) and the larger grid cell sizes used in a global simulation. Almost all LMs, however, are designed using leaf-scale or canopy-scale theories, and do not

contain an explicit length scale that modifies simulation characteristics for the size of the grid cell. Indeed many of these LMs are being used within regional modelling frameworks at resolutions that approximate this flux tower fetch. There are also very real benefits at the site scale. When direct observations of meteorological forcing and fluxes are co-located, we have an ability to quantify uncertainties, and errors in measurements of meteorological forcing are relatively small. Flux tower sites allow us to ascribe simulation errors to LMs in a way that is just not possible at gridded scales, where meteorological inputs to LMs

cannot be directly measured. Avoiding the 'garbage in, garbage out' problem, and ability to evaluate models at the time step size that they operate means that flux tower sites remain the data source that offers the best observational constraint for LM process evaluation.

## 2 Methodology

### 2.1 Flux tower data

In contrast to the 20 sites used in the PLUMBER experiment (Best et al., 2015), models completed simulations at 170 flux tower sites for PLUMBER2. While detailed explanations of the motivation, processing steps and quality control of site data are given by Ukkola et al. (2022), a brief overview is given here. At the time of processing, the aim was to maximise the number of sites that met variable availability and quality control requirements, as well as having open-access data.

FLUXNET2015, FLUXNET La Thuile Free-Fair-Use, and OzFlux collections were used as the starting point, and after processing with the FluxnetLSM package (Ukkola et al., 2017), it was ensured that sites: had reference (measurement) height, canopy height information and IGBP (International Geosphere–Biosphere Programme) vegetation type; whole years of data; and were not missing significant periods of key forcing variables (where gap-filling counted as missing), specifically incoming solar radiation (SWdown), air temperature (Tair), specific humidity (Qair) or precipitation (Rainf). Discerning thresholds in

these variables was clearly subjective, but involved consideration of the proportion of time series with measured data, length of gaps, coincidence between variables, and ubiquity of site type - see Ukkola et al. (2022) for detail.



Gap-filling (including allowing 100% synthesised data) of downwelling longwave radiation (LWdown) used the approach from Abramowitz et al. (2012). Surface air pressure (PSurf) was based on elevation and temperature, and ambient $CO_2$ was
based on global values (Ukkola et al., 2022).

Since most sites had no publicly available leaf area index (LAI) data, and none had time evolving LAI data, we specified a remotely-sensed LAI time series for each site to try to minimise differences between LMs. LMs that predict LAI would clearly not utilise this (Table 1). The LAI time series were derived from either MODIS (8-daily MCD15A2H product, 2002-2019) or
Copernicus Global Land Service (monthly, 1999-2017), with one of these chosen for each site based on a site-by-site analysis considering plausibility and some in-situ data, and provided as a single value for each time step of meteorological forcing. Time-varying LAI was provided for the time period covered by the remotely-sensed products and otherwise a climatology was constructed from all available years. We note that some LMs utilised this LAI estimate for a single vegetation type simulation and others partitioned it in a mixed vegetation type representation.


While all observational data contain measurement uncertainty, the issue of energy balance closure in flux tower data is particularly relevant in the context of this experiment. At a range of time scales, most sites do not obey the assumed equality of net radiation with the sum of latent heat flux, sensible heat flux and ground heat flux (see Wilson et al, 2002; Stoy et al, 2013; Mauder et al, 2020; Moderow et al, 2021). We therefore need to be careful attributing model-observation mismatch to
model error, since LMs are fundamentally constrained to conserve energy.

The simplest approach to dealing with this issue is to correct flux tower data for closure, where sufficient data exist. Energy-balance closure correction was part of the FLUXNET2015 release (the bulk of sites here) and we replicated this approach for sites from the other sources. Analyses below consider both raw and corrected latent and sensible heat fluxes, were conducted
only on flux time steps that were not gap-filled, and were also run separately filtered by time steps with wind speed above 2 ms$^{-1}$ so that potential concerns about measurement fidelity in low turbulence periods (typically night time) could be investigated.

The final forcing and evaluation files for all sites were ultimately produced in an updated version of ALMA NetCDF (Polcher
et al., 1998; 2000), with CF-NetCDF standard name attributes and CMIP equivalent names included where possible. A complete list of these variables, as well as those requested in LM output, are shown in Table S1. Table S2 shows a complete site list, with each site's included years, IGBP vegetation type, mean annual temperature (MAT) and precipitation (MAP), canopy height, reference height, elevation, chosen LAI representation, and references. Each tower site has a page on modelevaluation.org with more detail, including additional references, meta data, photographs and time series plots. We refer
readers to Ukkola et al. (2022) for a global map showing site locations. Site vegetation types and distribution in MAP-MAT space are shown in Fig. S1. Their location on a Budyko style dryness index versus evaporative fraction plot (Budyko, 1974;

Chen and Sivapalan, 2020) is shown in Fig. S2a. It is typically assumed that all sites will lie below 1 on the horizontal axis (i.e. evapotranspiration will be less than precipitation) and to the right of the 1-1 line (potential evapotranspiration > evapotranspiration), with drier, water limited sites close to 1 on the horizontal axis on the right hand side and wetter, energy
limited sites towards the bottom left hand side close to the 1-1 line.

This is however clearly not true for these site data. To understand why this was the case, we first examined cumulative precipitation at each site, compared to an in-situ based gridded precipitation product - REGEN (Contractor et al, 2020) - and identified those sites that appeared anomalous. Clearly there are many good reasons why site-based precipitation might
disagree with a gridded product, even if it were perfect. A subset of the sites were nevertheless identified as having precipitation data that was *a priori* not realistic, either because missing data had not been gap-filled (and was not flagged as missing, so precipitation flat lined), units had been reported incorrectly (e.g. US-SP1 appears to use inches rather than mm) or winter snowfall was apparently not included in precipitation totals (see Fig. S3). 16 sites were removed from the analysis as a result. These issues were unfortunately only identified after all modelling groups had completed their 170 site simulations, so the LM
analyses below are conducted on the remaining 154 sites.

While removing these sites did lessen the extent of the problem, it did not by any means solve it (see Fig. S2b - the same as Fig. S2a but with 154 instead of 170 sites). Next, we examined if using the entire time series for each site, instead of filtering out gap-filled time steps (Fig. S2a has gap-filled data removed) resulted in any qualitative change - it did not (see Fig. S2c).
Finally, we investigated whether using energy-balance corrected fluxes had an impact. Fig. S2d shows that it did indeed have a marked effect - but the proportion of sites where evapotranspiration exceeds precipitation increased.

Figures S2a-d reinforce how complicated the simulation task is for LMs, with around 30% of sites showing an average evapotranspiration exceeding average precipitation. Despite posing this as a data quality problem above, there are many sound,
physically plausible reasons for this, such as hillslope or preferential flow, irrigation or groundwater access by vegetation. Needless to say, most LMs will simply be unable to reproduce this behaviour since these inputs and processes are usually not included. We discuss more about this issue, its influence on results and implications for LM evaluation in the Results and Discussion.

## 2.2 Land model simulations

The model simulation protocol was broadly similar to that of PLUMBER. Groups ran mechanistic LMs offline in single-site mode (as opposed to gridded simulations), forced by standardised, locally observed meteorology for the 170 sites. Simulations were requested as "out-of-the-box", using default (usually vegetation type based) parameters for each site, as if the LMs were running a global simulation. Models used the IGBP vegetation type prescribed in each forcing file where possible, mapped to





the PFT schemes used by each model. In addition, site canopy height and reference height (measurement / lowest atmospheric

model layer height) were provided in each forcing file. No additional parameter information for sites was prescribed.

The rationale for this setup was to try to understand the degree of fidelity in flux prediction that LMs provide in a well-constructed global simulation (i.e. where meteorological forcing is as close as possible to being error-free and vegetation properties are appropriately represented by vegetation type), noting that different LMs had to adapt their representation

approaches in slightly different ways to achieve this (e.g. some use mixed vegetation types to describe a single location). While we might ideally additionally like to ensure that LMs used an appropriate soil type for each site, these are not universally measured or available for all sites, so LMs used default types from their default global soil type grid.

Different LMs require different periods of spin-up until model states reach an equilibrium, depending on whether or not they

include a dynamic carbon (and/or nitrogen / phosphorus) cycle(s), vegetation or stand dynamics, and how this is implemented. For models where a simple soil temperature and moisture spin-up is sufficient (e.g. if LAI is prescribed rather than predicted), we suggested that model spin-up use the site forcing file and repeatedly simulate *the entire* period, for at least 10 years of simulation, before beginning a simulation on the first year of site data.

For LMs with prognostic LAI and/or soil carbon(C), nitrogen (N), and phosphorous (P) pools, the process was more complicated. LM simulations were initialised with a spin-up routine resulting in equilibrium conditions of C stocks (and N and P if available) representing the year 1850. Climatic forcing for the spin-up came from the site eddy-covariance forcing file, which was continuously repeated. Atmospheric $CO_2$ and N deposition levels representing the year 1850 were set to 285 ppm and 0.79 kg N ha-1 yr-1, respectively. The transient phase covered the period 1851 to the year prior the first year in the site

data. LMs were forced with historical changes in atmospheric $CO_2$ and N deposition, continually recycling the meteorological inputs. The meteorological time series was repeated intact rather than in a randomised way, to avoid splitting of the observed meteorological years at the end of each calendar year. This of course does not accurately replicate the land use history of different sites, but in most cases, detailed site level histories were not available.

All models participating in PLUMBER2 are shown in Table 1. While some simulation setup information is included in the *Notes* column, more detailed information is available on the Model Output profile page for each set of simulations submitted to modelevaluation.org. While modelling groups were requested to report as many variables as possible from Table S1, the breadth of contributions were highly variable, so in an attempt to include all participants, analyses here focus on latent heat flux (Qle), sensible heat flux (Qh) and Net Ecosystem Exchange of $CO_2$ (NEE) only.




In addition to the LMs, two 'physical benchmarks' were also included, as per Best et al. (2015) - an implementation of a Manabe bucket model (Manabe, 1969) and a Penman-Monteith model (Monteith and Unsworth, 1990) with a reference stomatal resistance and unrestricted water availability.

**Table 1: Participating models. Land surface models that are developed as part of a coupled modelling system are denoted as 'climate' or 'weather' in their Notes, depending on their default application being climate projection or weather forecasting, despite simulations in this experiment being uncoupled/offline. In each case, leaf area index (LAI) is either prescribed, computed by the model itself, or not used (NA).**

| Model | Institution | LAI | Notes | Authors | References |
|---|---|---|---|---|---|
| BEPS | LBL, USA | Prescribed | Ecosystem model; v4.01 (https://github.com/JChen-UToronto/BEPS_hourly_site_4.02) | XL | Liu et al. (1997) |
| CABLE | UNSW Sydney / CLEX, Australia | Prescribed | Land surface model, climate; r8002; biophysics only, no C-N-P. | MdK, AU | Kowalczyk et al. (2006); Wang et al. (2011) |
| CABLE-POP | UWS / CSIRO, Australia | Prescribed | Land surface model, offline only. C-N cycle included. | JK | Haverd et al. (2013; 2016; 2018) |
| CHTESSEL (currently ECLand) | ECMWF, UK | Prescribed | Land surface model, weather. 3 simulation sets, forced locally and with ERA5. | DF, SB, GB | van den Hurk et al. (2000); Balsamo et al. (2009); Dutra et al. (2010); Boussetta et al. (2013) |
| CLM | NCAR, USA | Prescribed | Land surface model, climate; v5.0.34 | KO, DL | Lawrence et al. (2019) |
| ORCHIDEE2 | LSCE/IPSL | Computed | Land surface model, climate and $CO_2$ forcing. Model version used in CMIP6 (no C-N) | XW-F, CO, PP, NV | Krinner et al. (2005), |
| ORCHIDEE3 | LSCE/IPSL | Computed | Model version based on ORC2 but with Nitrogen cycle and C-N interactions | XW-F, CO,PP, NV | Vuichard et al.,( 2019) |
| JULES | Met Office, UK | Prescribed | Land surface model, weather and climate. | HR, MB | Best et al. (2011); Clark et al. (2011) |
| Manabe bucket | Met Office, UK | NA | Mechanistic benchmark | MB | Manabe (1969) |
| Penman Monteith PET | Met Office, UK | NA | Mechanistic benchmark; estimate of potential evapotranspiration (PET) | MB | Monteith & Unsworth (1990) |



| GFDL | NOAA GFDL, USA | Computed | Land surface model, climate | SM | Dunne et al. (2020); Shevliakova et al. (2023) |
|---|---|---|---|---|---|
| MATSIRO | U Tokyo, Japan | Prescribed | Land surface model, climate | TN, HK | MATSIRO6 Document Writing Team (2021) |
| STEMMUS - SCOPE | U Twente, Netherlands / Northwest Agriculture and Forestry U, China | Prescribed | Land surface model, offline only; R1.3.0 (https://github.com/EcoExtreML/STEMMUS_SCOPE) | YZ, YW, BS | Wang et al. (2021) |
| EntTBM | Yonsei U, Korea | Prescribed | Ecosystem model within the Earth System Modeling Framework (ESMF), coupled to the NASA Goddard Institute for Space Studies (GISS) GCM ModelE | YK, KC | Kim et al. (2015) |
| SDGVM | ORNL, USA / University of Sheffield, UK | Computed | Ecosystem model; carbon cycle model (https://bitbucket.org/walkeranthonyp/sdgvm/) | AW | Woodward et al. (1995); Woodward & Lomas (2004); Walker et al. (2017) |
| LPJ-GUESS | KIT, Germany | Computed | Ecosystem model; v4.0, Lund svn branch plumber, r8913; Qle is based on estimated evaporative demand and modelled soil water supply; Qh estimated from Rnet minus Qle (Note: LWnet component of Rnet and hence Qh during polar night at high latitude (lat > 64) sites is erroneous). | PA | Smith et al. (2014) |
| MuSICA | INRAE, France | Prescribed | Ecosystem model; Revision 710844c, veg params from Cable-POP, soil texture from soilgrids.org and soil hydraulic params from Montzka et al. (ESSD 2017) | MC, JO | Ogée et al. (2003); Gennaretti et al. (2020) |
| QUINCY | MPI BGC | Computed | Ecosystem model; | SC, SZ | Thum et al. (2019) |
| Noah-MP | NASA, U-Albany | Computed | V4.0.1; land surface model included in U.S. National Water Model and U.S. Unified Forecast System; run in NASA-LIS v7.2r (Kumar et al., 2006) ISRIC SoilGrids v2017 250m soil texture (Hengl et al. 2017); Vegetation fraction prescribed as annual maximum of NOAA-NCEP 1985-1989 AVHRR-based monthly climatology | CF | Niu et al. (2011); He et al. (2023) |
| Empirical: LSTM | University of Alabama | Prescribed | Empirical benchmark; three sites out-of-sample at a time. Two versions created - one to predict raw fluxes and on to predict energy-balance corrected fluxes | CB, JF, GN | |
| Empirical: RF | UNSW Sydney / CLEX Australia | Prescribed | Empirical benchmark; one site out-of-sample at a time. Two versions created - one to predict raw fluxes and on to predict energy-balance corrected fluxes. | SH, GA | |





| Empirical: clustering & regression | UNSW Sydney / CLEX Australia | NA | Empirical benchmark; one site out-of-sample at a time. Two versions created of 1lin and 3km27 - one to predict raw fluxes and on to predict energy-balance corrected fluxes | JCP, GA | Abramowitz (2012), Haughton et al. (2018) |
|---|---|---|---|---|---|


## 2.3 Empirical machine learning based benchmarks

The empirical models we use as benchmarks are also listed in Table 1. As suggested above, these are key to quantifying site predictability, and so setting benchmark levels of performance for LMs that reflect the varying difficulty or complexity of prediction at different sites, unknown issues with data quality at some sites and more broadly understanding the amount of information that LM inputs provide about fluxes. To do this meaningfully, all empirical models need to provide out-of-sample predictions. That is, every site simulation made by an empirical model here has not used that site's data to build/train the empirical model, and so cannot be overfitted to the particular characteristics or noise from the site. If the site is unusual, or its data is poor, the empirical models will provide a poor simulation, thus setting a lower benchmark of performance for the LMs.

A hierarchy of different empirical model approaches was used, firstly to avoid the eccentricities of a single technique or architecture, but more importantly, to give us an approximate equivalent level of performance for each LM within the hierarchy, as a way to rate its performance. Approximately from the simplest, with lowest performance expectations, to highest, these are:

- ***1lin***: a simple linear regression of each flux against downward shortwave radiation (SWdown), using half hourly data, training on 169 sites and predicting on one, repeated 170 times, as per Abramowitz (2012) and Best et al. (2015). Two versions were created - one trained to predict raw fluxes (1lin_raw) and one trained to predict energy-balance corrected fluxes (1lin_eb).
- ***2lin***: a multiple linear regression of each flux using SWdown and air temperature (Tair) as predictors, using half hourly data, training on 169 sites and predicting on one, as per Abramowitz (2012) and Best et al. (2015).
- ***3km27***: all site-timesteps of three predictors - SWdown, Tair and relative humidity (RH) - from 169 training sites are sorted into 27 clusters using k-means, and all site-timesteps in each cluster are used to establish multiple linear regression parameters against each flux for that cluster. Time steps at the prediction site are sorted into clusters based on proximity to cluster centres, and regression parameters for each cluster are then used to make predictions at the one remaining unseen, out-of-sample site, as per Abramowitz (2012) and Best et al. (2015). 27 clusters were chosen to approximately allow each predictor high, medium and low clusters: $3^3=27$. Two versions were created - one to predict raw fluxes (3km27_raw) and one to predict energy-balance corrected fluxes (3km27_eb).




- ***6km729***: As per 3km27, but using six predictors - SWdown, Tair, RH, Wind, Precip, LWdown (see Table S1 for variable definitions) - and 729 k-means clusters, training on 169 sites and predicting on one, similar to Haughton et al. (2018). 729 clusters were chosen to approximately allow each predictor high, medium and low clusters: $3^6$=729;

- ***6km729lag***: As per 6km729, but with lagged Precip and Tair as additional predictors. These took the form of six additional predictors: mean Precip and Tair from the previous 1-7 days, 8-30 days and 31-90 days. Training on 169 sites and predicting on one, similar to Haughton et al. (2018);

- ***RF***: A Random Forest model with Tair, SWdown, LWdown, Qair, Psurf, Wind, RH, CO2air, VPD, and LAI as predictors. These predictor variables are listed in order of variable importance. While Precip was originally included, it actually offered negative variable importance - suggesting that including Precip reliably *degraded* the empirical prediction out-of-sample. Training was on 169 sites and predicting on one out of sample, repeated 170 times. As a nominally more sophisticated empirical model than the cluster+regression approaches above, RF offers a lower bound estimate of predictability of fluxes from instantaneous conditions (no lags). Two versions were created - one to predict raw fluxes (RF_raw) and one to predict energy-balance corrected fluxes (RF_eb).

- ***LSTM***: A Long Short-Term Memory model given as much information as the LMs. Two types of input features were used for training: dynamic features (that change for each time step - CO2air, LWdown, Precip, Psurf, Qair, RH, SWdown, Tair, VPD, Wind and LAI - and static site attributes that are constant per site (MAT, range of annual MAT, MAP, mean LAI, range of annual LAI, elevation, canopy height, reference height, latitude, mean SWdown, PET and IGBP vegetation type). Training was on 167 sites and prediction was on the three remaining sites (randomly chosen), repeated to make out-of-sample predictions at all sites. A single LSTM was used to predict Qle, Qh and NEE simultaneously, to account for the fact that the three fluxes are all components of a highly coupled system. The LSTM provides a lower bound estimate of predictability of fluxes using both instantaneous and meteorological conditions and internal states based on them - a proxy for LM states. Two versions were created - one to predict raw fluxes (LSTM_raw) and one to predict energy-balance corrected fluxes (LSTM_eb).

As noted above, all predictions made by empirical / machine learning models are out-of-sample at the site level which means that all of the data from the site being predicted is excluded from the training set for all the empirical models above. We also note that the 1lin, 3km27m, LSTM, and RF models made two separate sets of predictions of Qle and Qh: one when trained on the raw fluxes and one trained on the energy balance corrected fluxes.

## 2.4 Analyses

The dimensionality and complexity of the data resulting from this experiment obviously presents many options to interrogate the performance of LMs. Our analysis here focuses on a relatively high-level overview without any intention to be comprehensive - we anticipate that analysis of these simulations will extend well beyond this paper and will take some time.



Here we consider a few canonical approaches: mean flux variability, variable ratios such as evaporative fraction and water use efficiency (although using NEE rather than GPP), separation by vegetation type and viewing results in a Budyko curve context.

Where possible, we try to (a) be summative, avoiding using any one metric or site at a time, and instead try to present results collectively across all site and an independent metric set, and (b) present LM performance *relative to a benchmark*, rather than

using raw metric values. To better understand what we mean by this, let's first look at the metrics we use, shown in Table 2.

**Table 2: The independent set of metrics used to assess aggregate performance. Metrics are calculated on all time steps ($i=1,..,n$) of observations (O) and each model (M) separately for each site. $B$ is the number of bins used for density estimation, in this case 1024.**

| Mean Bias Error (MBE) | $\frac{\sum_{i=1}^{n}(M_i - O_i)}{n}$ |
|---|---|
| Standard Deviation difference (SD) | $\left\| 1 - \frac{\sqrt{\frac{\sum_{i=1}^{n}(M_i - \bar{M})^2}{n-1}}}{\sqrt{\frac{\sum_{i=1}^{n}(O_i - \bar{O})^2}{n-1}}} \right\|$ |
| Correlation Coefficient (r) | $1 - \left[ \frac{n\sum_{i=1}^{n}(O_i M_i) - (\sum_{i=1}^{n} O_i \sum_{i=1}^{n} M_i)}{\sqrt{\left(n\sum_{i=1}^{n} O_i^2 - (\sum_{i=1}^{n} O_i)^2\right)\left(n\sum_{i=1}^{n} M_i^2 - (\sum_{i=1}^{n} M_i)^2\right)}} \right]$ |
| Normalised Mean Error (NME) | $\frac{\sum_{i=1}^{n}\|M_i - O_i\|}{\sum_{i=1}^{n}\|\bar{O}_i - O_i\|}$ |
| 5th percentile difference (5th) | $\|M_5 - O_5\|$ |
| 95th percentile difference (95th) | $\|M_{95} - O_{95}\|$ |
| Density estimate overlap percentage (PDF) | $100 - \left[ \frac{range(M,O)}{B} \cdot \left(\sum_{b=1}^{B} min(M_b, O_b)\right) \cdot 100 \right]$ |


This set of metrics is independent, in the sense that for a given observational time series, a change can be made to the model time series that will affect any one of these metrics without affecting the others. This is not true, for example, of RMSE and correlation. Metrics are calculated separately for each model at each site.

Next, we combine the information we get from this independent metric set into a single performance measure. To do this, we first set a reference group of benchmark empirical models, and compare all LMs to this reference group. Suppose we wish to compare a given LM against 1lin, 3km27 and LSTM, for example. Then, for each metric (*m*), at each site and and for each



variable, we have metric values for the LM, 1lin, 3km27 and LSTM. We then define the normalised metric value (NMV) for this LM at this site, for this variable and metric, in one of two ways.


First, staying with the broader approach used by Best et al. (2015), we define *dependent* NMV by including the LM in the metric range being normalised:

$$dNMV_{LM} = \frac{m_{LM} - min(m_{1lin}, m_{3km27}, m_{LSTM}, m_{LM})}{max(m_{1lin}, m_{3km27}, m_{LSTM}, m_{LM}) - min(m_{1lin}, m_{3km27}, m_{LSTM}, m_{LM})} \tag{1}$$


So dependent NMV simply denotes where in the metric range of these 4 models the LM was, scaled to be between 0 and 1, with lower values representing better performance. This allows us to average NMV over metrics, sites, variables, vegetation types or other groupings to get an aggregate indication of performance. This is similar to the approach used in Best et al. (2015), except using the average of normalised metric values, rather than discrete metric value ranks. Using metric value ranks effectively forces even spacing of metric values, which can give misleading results, particularly in cases when some models' metric values are clustered.

The second approach, using an *independent* NMV, defines the normalised metric range using only the reference benchmark models:

$$iNMV_{LM} = \frac{m_{LM} - min(m_{1lin}, m_{3km27}, m_{LSTM})}{max(m_{1lin}, m_{3km27}, m_{LSTM}) - min(m_{1lin}, m_{3km27}, m_{LSTM})} \tag{2}$$

Using this second approach, *iNMV* allows us to define lower and upper performance expectations to be independent of the LM being assessed. In this case we might expect that 1lin will typically have a value of 1 and LSTM 0, and the LM, if its performance lies between these two, will have a value somewhere in this interval. It also allows the LM to score a much lower value than zero, if it performs much better than the empirical models, and conversely, a value much larger than 1 if it is much worse.

To illustrate why such a detailed approach to analysis is necessary, we now briefly show why some common heuristic measures of performance are inadequate. Figures S4, S5 and S6 (supplementary material) show the performance results of the 1lin model at the US-Me2 site, examining latent heat flux, sensible heat flux and NEE in three different common graphical performance measures. These are: the average diurnal cycle of NEE, shown for different seasons (Fig. S4); a smoothed time series of Qh (Fig. S5); and the average monthly values of Qle showing the evaporative seasonal cycle (Fig. S6). In most contexts, if these blue curves were plots of a LM's performance, the reader would accept this as qualitative or even quantitative evidence of excellent LM performance. Yet these represent perhaps the simplest possible model - a simple linear regression against



shortwave, out-of-sample (trained on other sites only). They illustrate just how much site variability can be simply driven by instantaneous shortwave radiation, and that visual closeness of curves, and an ability to capture seasonal variability, diurnal variability and even interannual variability should not *a priori* be accepted as evidence of good model performance.

As noted above, all analyses were filtered to exclude time steps at each site where observational flux data was flagged as missing or gap-filled. Analyses were half-hourly or hourly, depending on the reported time step size at each site, except for models that only reported monthly outputs, which were then analysed with monthly averages. All data management and analyses were conducted through https://modelevaluation.org  (see Abramowitz, 2012), and can be repeated there. The analysis codebase used for PLUMBER2 within https://modelevaluation.org is available at https://gitlab.com/modelevaluation/me.org-
r-library.

## 3. Results

Figure 1 shows the average latent heat flux (Qle) versus sensible heat flux (Qh), averaged across all sites for participating models that reported both of these variables. Dashed lines show a proxy for observed available energy (around 69 Wm$^{-2}$, defined as Qle+Qh, assuming mean ground heat flux on longer time scales is zero) and observed Bowen ratio (around 0.7).
Perhaps unsurprisingly, models differ most in their partitioning of surface energy (spread along the available energy axis) rather than amount of available energy (spread along the Bowen ratio axis), supporting previous findings (see Haughton et al, 2016). Those LMs that do not operate in a coupled modelling system (i.e. coupled to an atmospheric model; EntTBM, LPJ-GUESS, MuSICA, QUINCY, STEMMUS-SCOPE) also appear to have a much broader spread of estimates than those used in coupled models (they are furthest from the observed Bowen ratio in Fig. 1).





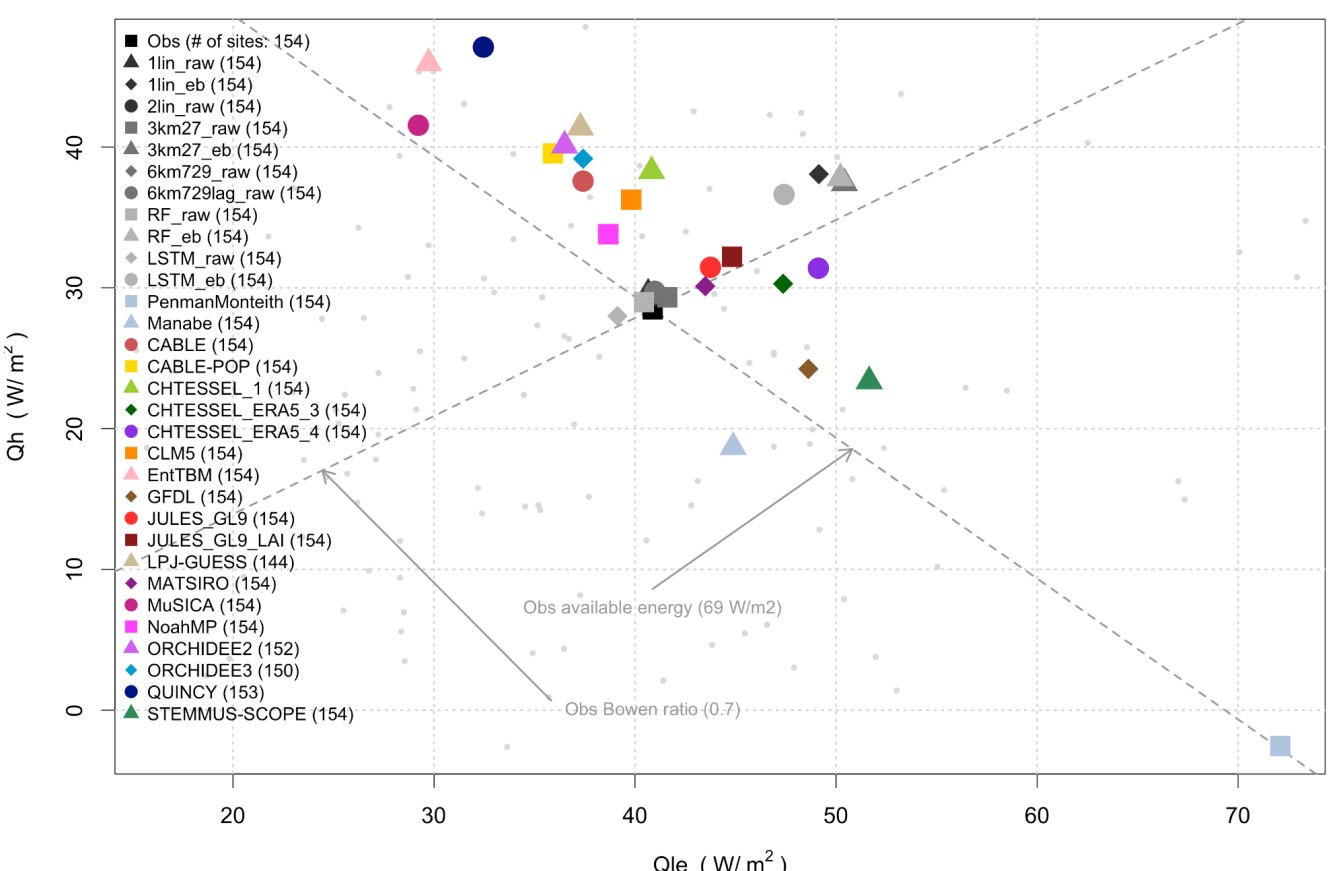

**Figure 1: Average latent heat flux (Qle) versus average sensible heat flux (Qh) averaged over 154 sites, shown for all models that submitted both quantities. Dashed lines show observed constant values of average available energy (Qle+Qh) and average Bowen ratio (Qh/Qle) across the sites, using raw (as opposed to energy balance corrected) flux data. Light grey dots in the background represent individual site averages.**

When averaged across all sites, the LMs do not appear to show any clear systematic bias in energy partitioning relative to observations across the ensemble. Note that in Fig. 1 the observations *do not* have the Fluxnet2015 energy-balance correction applied (the equivalent figure using energy balance corrected fluxes is shown in Fig. S7a). Aside from showing a little more available energy (their mean is slightly offset from the observed available energy line, by less than 10%), the LMs are relatively evenly spread around the observational Bowen ratio. This lends little support to an argument of systematic observational bias in the partitioning of available energy. Perhaps unsurprisingly, observations, once energy-balance corrected, lie in amongst the empirical models trained to predict energy-balance corrected fluxes (the cluster of grey points with higher available energy in





Fig. 1 - see Fig. S7a). The average Bowen ratio increases slightly to 0.73 instead of 0.7 with energy-balance correction. Perhaps more interesting is that the corrected version of flux observations contains an average of 16 Wm$^{-2}$ additional energy across
these sites, about a 23% increase, and that this value sits much further outside the spread of modelled estimates of available energy than the observed value in Fig. 1. So in this metric at least (and indeed in more below), the LMs' performance is not improved when energy-balance corrected flux data is used. In the discussion section we consider whether results in this paper might raise some doubt about whether the approach to energy balance correction used in Fluxnet2015 is ideal - further results below are somewhat mixed. We present results comparing with raw fluxes in the main part of this manuscript, with
comparisons against energy-balance corrected data, where they qualitatively differ, shown in Supplementary Material. Similarly, when we filter analyses to only include time steps with wind speed above 2 ms$^{-1}$ (Fig. S7b), the scatter of models in Fig. 1 changes surprisingly little.

## Average Qle vs NEE over all sites

**Figure 2: Average latent heat flux (Qle) versus average net ecosystem exchange of CO2 (NEE) averaged over 154 sites, shown for all**
**models that submitted both quantities. The observed value is shown in black with crosshairs. Light grey dots in the background**




**represent individual observed site averages, with the linear fit between them shown in bold dashed grey. Regression lines are also shown for LMs showing a stronger fit than in the observed case ($R^2$=0.19).**

Figure 2 is similar to Fig. 1, but shows average latent heat flux (Qle) versus Net Ecosystem Exchange of $CO_2$ (NEE) for those

LMs that reported both variables. Given the expectation that NEE is likely to be strongly dependent on site history, and that we could not reliably include this information in the modelling protocol or account for it in this plot, there is no a priori reason to expect a clear relationship here. While we might broadly expect increasing carbon uptake with increasing Qle, as shown by the observed regression line in Fig. 2, the fit is relatively weak ($R^2$ is 0.19). LM regressions are shown where their fit has higher $R^2$ than observed, although we note that aside from ORCHIDEE2, CABLE-POP and NoahMP, only empirical models

meet this criterion (since they effectively act as data smoothers).

With the exception of Noah-MP, STEMMUS-SCOPE and some empirical models, all LMs predict less net carbon uptake than is observed. This may well be because the models were run without any site history. That is, the simulated ecosystems were closer to equilibrium than those in the real world. In equilibrium, vegetation and soil carbon stocks are high and thus respiration

is also higher as it is generally simulated as a function of carbon stocks. Ecosystem models predict the least carbon uptake but a large range in Qle values (MuSICA, LPJ-GUESS, QUINCY, SDGVM). The equivalent plot with energy balance corrected Qle values (not shown) simply moves the 'observed' black square (currently below RF_raw) to the right, once again sitting amongst 1lin_eb, 3km27_eb, RF_eb and LSTM_eb.

We also note that while LMs' spread might well be because of a lack of site history information, the empirical models show that missing this information does not actually reduce NEE predictability to a large degree (all empirical models are within 0.35 μmol/m²/s of observations). The empirical models also do not have any site history, and indeed in most cases, do not even use any estimate of LAI. They are trained only at other sites, so they cannot infer any site history information from the meteorology-flux relationship. Despite this, they cluster quite tightly around the observations in Fig. 2, whether predicting raw

Qle (cluster of grey points in the crosshairs) or energy-balance corrected Qle (cluster of grey points to the right of this). They all suggest a net uptake of C across these sites, within a narrow range spread around the observations.







**Figure 3: The average performance across all 154 sites and 7 metrics for Qh, Qle and NEE (lower is better). Average performance**
**is the mean of dependent normalised metric values (dNMV) within the range of metric values across models being compared in each**
**panel (4 in total, the LM (blue) - shown in plot title - and three reference benchmarks: 1lin_raw (red), 3km27_raw (yellow) and**
**LSTM_raw (green)). The first 10 panels (faded) show empirical or physical benchmark models.**

Figure 3 shows modified 'PLUMBER plots', similar to Best et al. (2015), but here using the average of the dependent
normalised metric values (dNMV) in the range of metric values across the four models being compared in each panel (one
LM, 1lin_raw, 3km27_raw and LSTM_raw). This is as opposed to the average rank of metric values used in Best et al. (2015),
which can distort results when metric values are clustered. Each panel in Fig. 3 shows the model in the panel title in blue, with
benchmark empirical models in red (1lin_raw), yellow (3km27_raw) and green (LSTM_raw). Lower values represent better
performance. LMs are shown alphabetically, with the first 10 panels, faded, showing the remaining empirical models and
physical benchmarks against these three benchmark models.



The out-of-sample LSTM_raw on average performs best across all fluxes, for these sites and metrics. The performance of LMs is highly variable, with half of them performing better than the 3km27_raw model for Qle, 15 of 18 worse than the out-of-sample simple linear regression (1lin_raw) for Qh, and NEE typically between the 1lin_raw and 3km27_raw levels of performance (12 of 14 LM variants). Overall, it's clear that LMs tend to perform better against the benchmarks for Qle and NEE than Qh, typically falling within the range of these three benchmarks for Qle and NEE. CLM5, MATSIRO and NoahMP are the only LMs with Qh metrics within this range. The LMs falling outside the benchmark dNMV ranges for Qle and NEE are a mixture of LSMs and ecosystem models. The equivalent plot using energy balance corrected Qle and Qh observations is shown in Fig. S8a. The performance of the LMs against the benchmarks remains remarkably similar, with some LMs slightly better and others slightly worse against corrected data. Filtering for higher wind speed time steps (Fig. S8b, using raw flux data) also appears to make no qualitative difference, if anything making LM performance worse relative to these empirical benchmarks.

When we look at the same set of figures using *independent* normalised metric value (iNMV) instead of dependent (dNMV), the picture is very different (Fig. 4). Recall that iNMV sets the normalised metric range (0,1) based on the three reference out-of-sample empirical models (1lin, 3km27 and LSTM) only, rather than these three *and* the LM, and then compares the LM to this range. For example, if the three reference empirical models have a mean bias in Qle of 35Wm$^{-2}$, 28Wm$^{-2}$, and 25Wm$^{-2}$ (a range of 10Wm$^{-2}$), and the LM has a bias of only 10Wm$^{-2}$, the iNMV of the reference models is 1, 0.3, and 0, respectively, and the LM has iNMV of -1.5 (remembering that lower is better). Alternatively, if the LM has a bias of 50Wm$^{-2}$, its iNMV would be 2.5. So iNMV values are not constrained to be in the unit interval, as they are for dNMV.

Figure 4 shows the same data as Fig. 3, but using iNMV instead of dNMV. The values of iNMV for the three reference models are now identical across all LM panels, so the values of iNMV for each LM are directly comparable. Note however that the vertical axis scale is different in each panel, so we can see the range for each LM. LM performance in iNMV clearly looks a lot worse. It tells us that when LMs perform worse than the out-of-sample linear response to shortwave, 1lin, they often perform a lot worse (at least a lot worse relative to the range between 1lin and LSTM_raw). While some LMs (CABLE, CABLE-POP, CHTESSEL, CLM, JULES, NoahMP and ORCHIDEE) perform within the range of the three empirical models for some variables, averaged over all variables, no LM outperforms the out-of-sample linear regression against SWdown. This is a sobering result. LM performance is particularly poor relative to the benchmarks for Qh with no models within the range of the benchmarks (compared to 40% of them for Qle and 29% for NEE).

Equivalent plots to Fig. 4 using energy-balance corrected fluxes (Fig. S8c) and time steps with wind speed > 2ms$^{-2}$ (Fig. S8d) are shown in supplementary material. Again, LM performance appears remarkably similar despite the significant changes made with the target energy-balance corrected data (Fig. 1). It remains true that no LMs outperform the 1lin averaged over all





fluxes. Note that in this comparison, where energy-balance corrected data are the reference target, the versions of empirical models trained for this target are used for comparison (i.e. 1lin_eb, 3km27_eb and LSTM_eb).

We also note that despite this result, some LMs do perform better than the empirical benchmarks for a subset of the metrics in Table 2, for some variables. Figs S7e - S7k are versions of Fig. 4 constructed with only one metric at a time. LMs tend to
perform better in the 5th percentile and PDF overlap metrics, and worst in temporal correlation and NME. It's also apparent that RF, 6km729 and 6km729lag all outperform the LSTM in quite a few of these metrics. Despite this, we did not investigate alternatives to the LSTM as the high level empirical benchmark.

**Figure 4: As per Fig. 3, but using the average of independent normalised metric values (iNMV) defined by the range of metric values across the three reference models (1lin_raw, 3km27_raw and LSTM_raw). Note that different panels have different y-axes.**



Figure 5 shows how closely LMs approximate the density of observed values of Qle, Qh and NEE. The thick black line shows the observed density estimate of each variable. The thinner lines are the modelled density estimate minus the observed density
estimate, and so show which magnitudes of fluxes are simulated too often (positive) or not often enough (negative). The vertical axis is the same for both observed density and LM density error. Most obvious in these plots is that the simpler empirical models (1lin, 2lin, and to some degree 3km27) tend to estimate too many positive values of NEE between 1-2 µmol/m²/s, Qh values around -20 to -30 Wm⁻² and Qle values around 5-10Wm⁻². With the exception of CABLE, EntTBM and QUINCY, most LMs underpredict near zero Qle values. JULES appears to simulate near zero Qh values far too often, as
CABLE does near zero Qle values, and ORCHIDEE and CHTESSEL near zero NEE values. The equivalent figure using energy balance corrected Qle and Qh fluxes is shown in Fig. S8, and is remarkably similar to Fig. 5. The same plot filtered for wind speed was also qualitatively very similar, and so is not shown here.



**Figure 5: Error in the density of values of Qle, Qh and NEE relative to observations collated across all sites. The thick black lines show the density of observed values of each variable, and thinner lines error in each model's variable density, on the same vertical axis. Horizontal axes are restricted to the region of highest density difference**.

Figure 6 shows boxplots of error in the average energy evaporative fraction (EF) across the 154 sites, shown separately for
each participating model. Energy evaporative fraction is defined using average flux values at each site: Qle / (Qle+Qh). The equivalent plots using energy balance corrected data and data filtered for wind speed are almost indistinguishable from Fig. 6, and so have not been included. In keeping with what we saw in Fig. 1, the mechanistic benchmarks and ecosystem models show the largest deviation from site observations, and empirical approaches are reliably zero-centred despite having no explicit mechanism to constrain the ratio between Qle and Qh. Once again, there does not appear to be any obvious reason to suspect
a bias in partitioning in observations - some LMs (6) show a high EF bias, and others (11) a low bias.

An equivalent version of this figure showing water evaporative fraction - Qle / Rainf - is shown in Fig. S10a and Fig. S10b (in supplementary material), using raw and energy-balance corrected fluxes, respectively. Once again, models are well scattered about the zero error line when raw fluxes are used, and almost all appear strongly negatively biased when compared to the
energy balance corrected fluxes. The equivalent plots using wind speed filtered data are qualitatively the same as Figures S10a and S10b, and so are not included here.



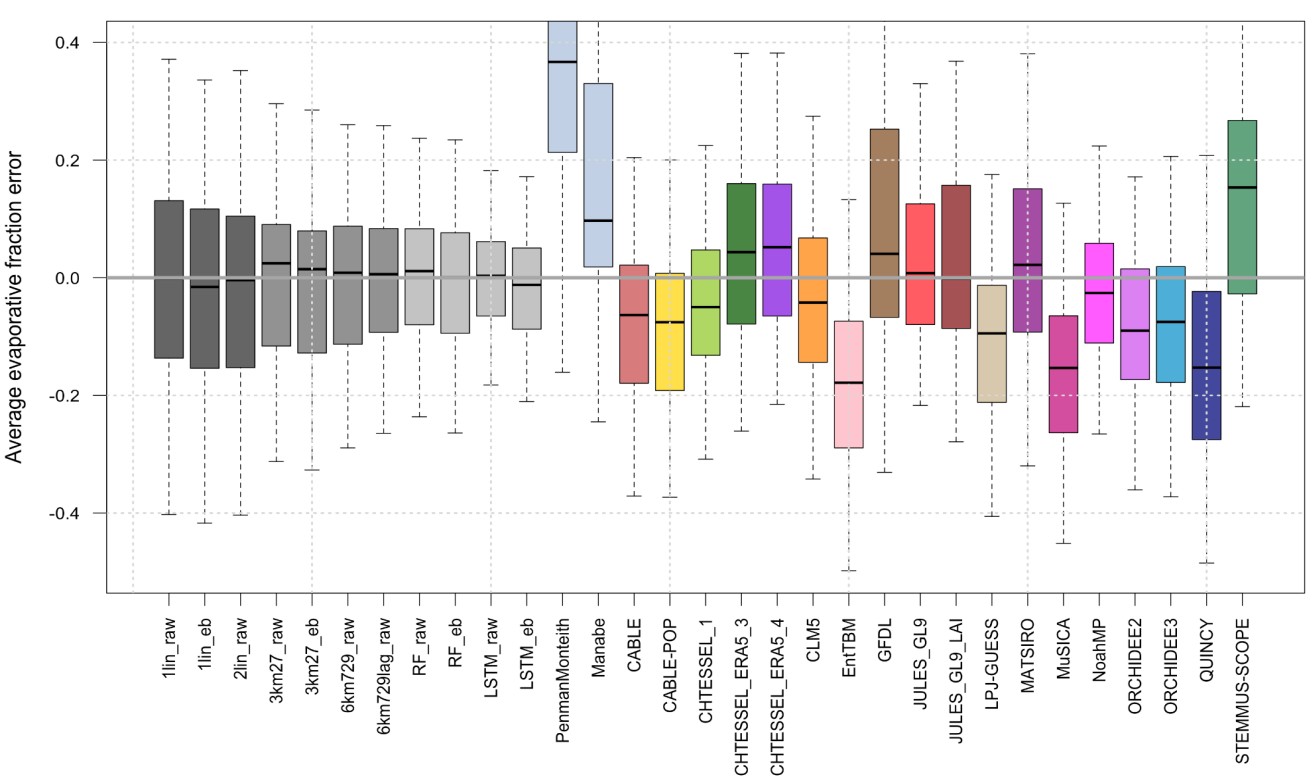

**Figure 6: Box plots of error in site mean energy evaporative fraction (Qle/(Qle+Qh)) over all sites, shown separately for each model, using raw flux data across 154 sites.**


Figure 7 is similar to Fig. 6, but shows error in water-use efficiency (NEE/Qle), expressed in units of $\mu$mol of carbon gained per gram of water (left vertical axis) and error as a percentage of observed WUE (right vertical axis), with the heavy pink dashed lines representing +/- 100%. It shows that almost all LMs underestimate WUE, typically by about 50%, presumably related to the broad under-prediction of NEE by LMs evident in Fig. 2. At the other end of the spectrum, NoahMP shows a

very high WUE bias, consistent with its overprediction of C uptake in Fig. 2 (due to a high dynamically predicted LAI). The empirical models, without any explicit constraint on the ratios of predicted variables (they are predicted independently), are better spread around observed values. Note that this statement applies equally to those empirical models trained on raw flux tower data and those trained on energy balance corrected data. The equivalent plot using energy balance corrected Qle data is shown in Fig. S10, and looks qualitatively similar to Fig. 7. For this metric, there are no discernable differences in performance

across types of LM.



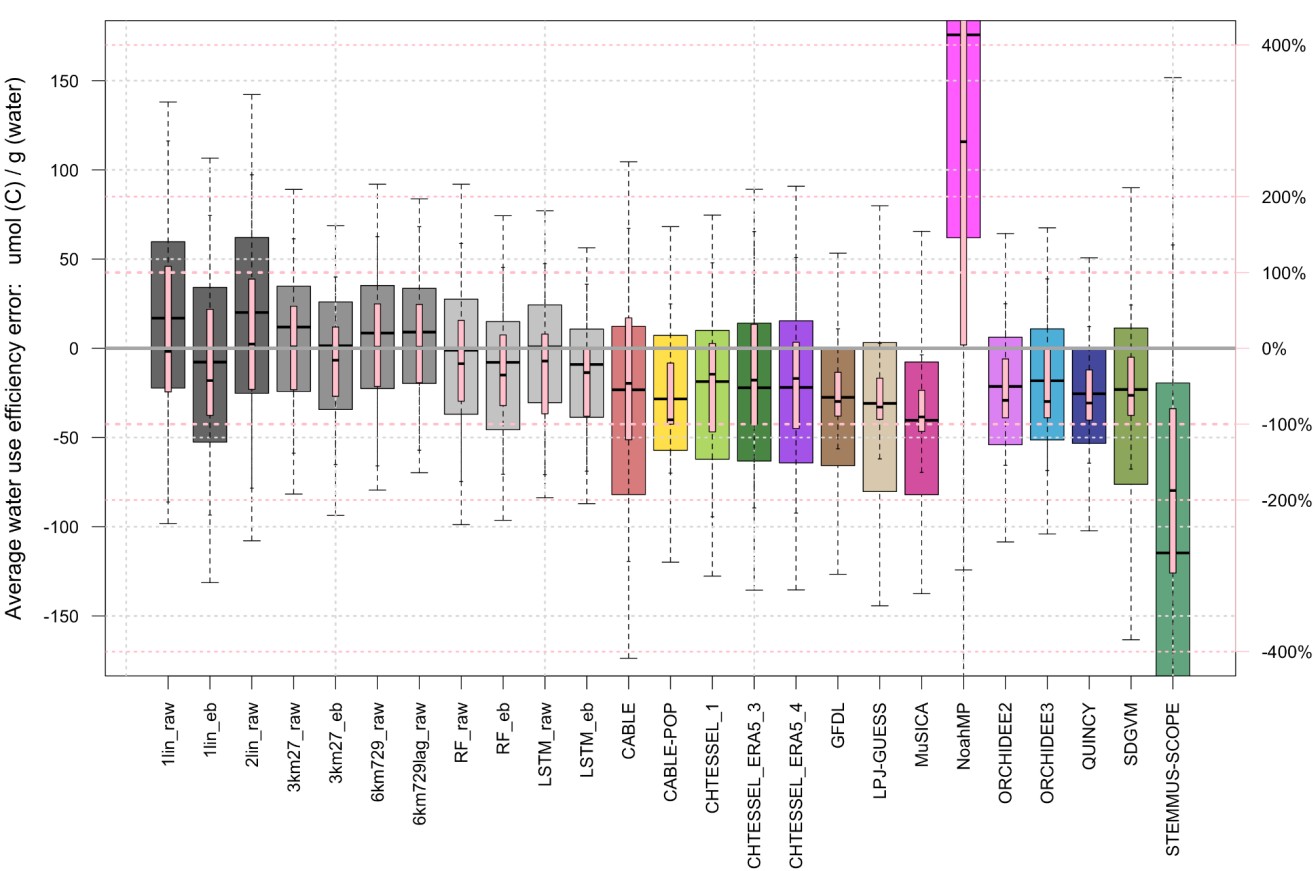

**Figure 7: Box plots of error in site mean ecosystem water use efficiency (-NEE/Qle) over all sites, shown separately for each model. WUE error is expressed both in units of umol of C gained per gram of water lost (left vertical axis, grey and multicoloured box plots) and error percentage of observed WUE (right vertical axis, pink box plots), with the heavy pink lines representing +/- 100%.**

We now focus on utilising the better performing empirical models as a benchmark for the mechanistic models. While they are the best performing models in this collection, they provide a *lower bound* estimate of predictability of fluxes at each site, since we can almost certainly produce better empirical models. The discrepancy between our best performing out-of-sample empirical models and a given mechanistic model defines an amount by which we *know* that the mechanistic model can improve by. This also allows us to define model performance in a way that accounts for site complexity / peculiarity / predictability, as well as observational errors particular to each site, and avoids some misleading statistics, like large RMSE values at sites that simply have larger fluxes. For this purpose, we use one of the best performing empirical models as the reference model, LSTM_raw.



First, in Fig. 8, we look at the discrepancy in iNMV between each mechanistic model and LSTM_raw for latent heat flux predictions. Results are shown separately for each IGBP vegetation type, and only the interquartile range and median estimates are shown for each boxplot. Note that these are the observed vegetation types for each site, and that some LMs with dynamic vegetation might represent these sites differently. Values below zero show that the LM performed better than the three

benchmark empirical models (1lin, 3km27, LSTM). Values between zero and one mean that the LM performed within the range of the benchmark models (shaded grey background), and above one means that the LM was worse than 1lin. The average of all vegetation types for each mechanistic model is shown by the dark grey horizontal line and number in the lower section of each panel, with the zero line in light grey. Each box plot represents the difference in independent normalised metric values across all metrics in Table 2.


Figure S12a shows the equivalent to Fig. 8 using energy-balance corrected fluxes, and Figures S12b and S12c show the same for sensible heat flux, with raw and energy-balance corrected fluxes, respectively. As is apparent in Fig. 3, the LSTM comfortably outperforms the LMs in general. And while some LMs show improved performance using energy balance corrected data, others show degradation, although more appear to improve, and by larger margins.


Figure 9 shows the same information for NEE predictions. While results remain highly variable across LMs, there is evidence that some ecosystem models that appeared worse in other fluxes are better at carbon flux prediction (e.g. LPJ-GUESS, QUINCY, SDGVM), competing with some of the more complex LSMs. Perhaps surprisingly, there appears to be no relationship between those LMs that performed within the 0-1 iNMV range (CABLE, CABLE-POP, CHTESSEL_1,

CHTESSEL_ERA5_3, CHTESSEL_ERA5_4, GFDL, ORCHIDEE2, ORCHIDEE3, QUINCY) and whether LAI was prescribed or computed dynamically (4 of these has computed LAI). Those outside the 0-1 iNMV range were also mixed in the LAI simulation approach (see Table 1).





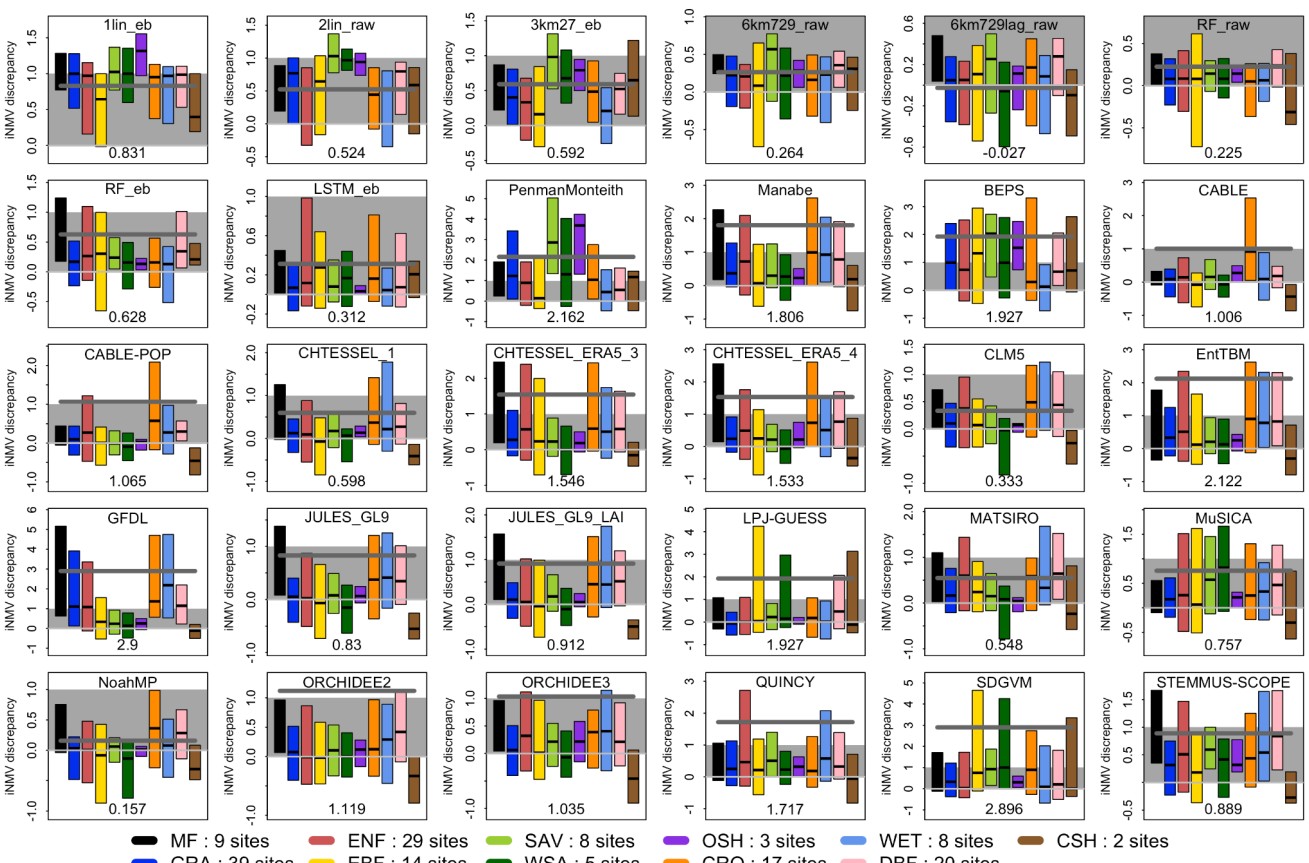

**Figure 8: Normalised metric discrepancy between each mechanistic model and LSTM_raw for latent heat flux (Qle), with separate inter-quartile boxes for each vegetation type, using raw fluxes. Mean model discrepancy is shown by the dark grey line and text at the bottom of each panel, with reference empirical model range [0,1] shaded in grey. Vegetation types are: Mixed Forest (MF); Grassland (GRA); Evergreen Needleleaf (ENF); Evergreen Broadleaf (EBF); Savanna (SAV); Woody Savanna (WSA); Open Shrubland (OSH); Cropland (CRO); Wetland (WET); Deciduous Broadleaf (DBF); Closed Shrubland (CSH). Lower scores are better.**





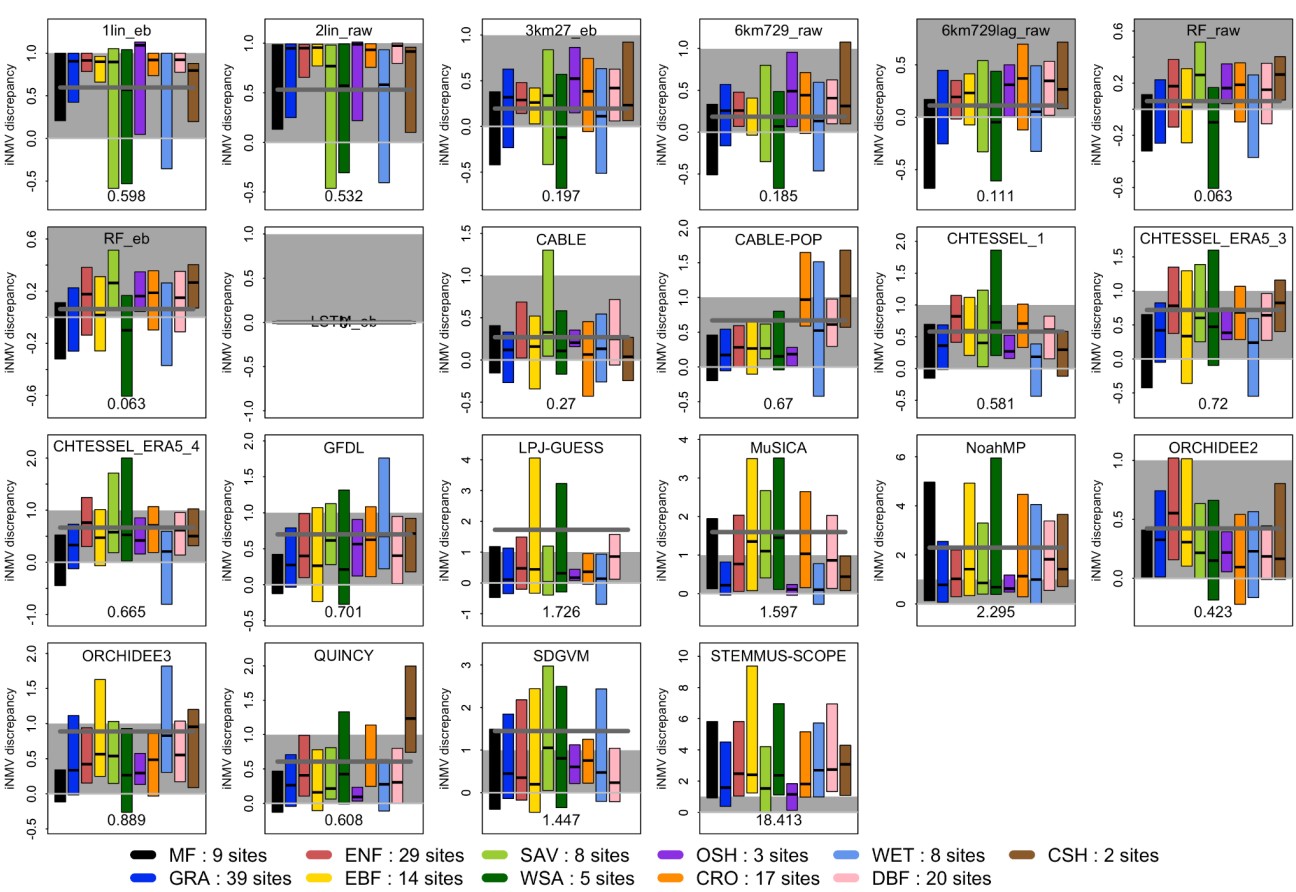

**Figure 9: As per Fig. 8, but for Net Ecosystem Exchange of CO2 (NEE), for those LMs that reported NEE. Lower scores are better.**


Given that we've seen that mechanistic LMs are broadly unable to meet the benchmarks provided by their out-of-sample empirical counterparts, we now briefly investigate whether there are obvious signs that biases are shared across the mechanistic models.

Figure 10 shows the same information as Fig. 8, but this time sorted by vegetation type, with each model shown as a separate interquartile box plot. This time the average of all models is shown by the dark grey bar, and the zero line in light grey. There are clearly variations across vegetation types, and while mean LM performance is worst for open shrubland (OSH), evergreen broadleaf forest (EBF) and mixed forest (MF), results across different LMs vary significantly. Overall, LM performance appears better for grass-dominated vegetation types (grassland and savannas) than tree ecosystems. The equivalent plots for

Qh (Fig. S13b), using energy-balance-corrected data (Fig. S13a for Qle; S13c for Qh) and NEE (Fig. S13d) seem to confirm





the lack of clear differentiation by vegetation type, and do not broadly show particular vegetation types as consistently anomalous. While some of the LM means (dark grey line) appear to change markedly for Qle after energy-balance correction (most notably for grassland sites), note that this seems largely because of significant changes to outlier LMs, rather than a change in aggregate behaviour. This is definitely less of a change for Qh as a result of energy-balance correction. Also note

that in all of these figures, the LM mean is often well above most of the 25th-75th percentile box plots. This simply reinforces that point made above that when LMs are worse than the reference benchmarks, they are often much worse (the smallest and largest 25% of values do not contribute to these box plots, obviously).

**Figure 10: Independent normalised metric discrepancy between each model and LSTM_raw for latent heat flux (Qle), as per Fig. 7,**
**but sorted by vegetation type. The average of all LMs for each vegetation type is shown by the bold dark grey line, and the zero line is in light grey. Lower scores are better.**





Figure 11 shows the same data - iNMV improvement offered by the LSTM over models - on a per-site basis, with the median difference for all LMs plotted (shown by colours). Each site's location is shown on axes of observed water evaporative fraction versus dryness index, as per Figures S2a-d. Note that the location of the 1-1 line relative to the sites is very much dependent upon our estimate of potential evapotranspiration (PET), which here is given by the Penman-Monteith model described above, so it's entirely plausible that a different estimate would see all sites (with the exception of US-Bkg) lying to the right of the 1-1 line. We might also wish to plot a curve on this figure illustrating the Budyko hypothesis (Budyko, 1974; although there is no single accepted derivation of an equation that describes the asymptotic behaviour it suggests; see Sposito, 2017; Mianabadi, 2019), but the spread of sites should make it clear why this is not particularly useful. Many sites have an evaporative fraction above 1. This reinforces that the conceptual idealisation of the Budyko hypothesis applies only at very large spatial scales and/or in idealised circumstances of water availability. Irrigation, or landscape features like topography/hillslope, sub surface bedrock bathymetry or groundwater can mean it is entirely physically reasonable for a location to exhibit an evaporative fraction above 1, as around 30% of these sites do. These factors are likely to still be relevant at scales of 10s of kilometres, so it seems unreasonable to suggest these effects are not also relevant for gridded simulations.

Of the sites in Fig. 11 with evaporative fraction greater than 1, only one is irrigated (ES-ES2). Hillslope factors are quite plausibly important in four others (CN-Dan, DK-ZaH, US-SRG, US-SRM). One is affected by fire prior to the measurement period, which might mean that accumulated water was available (US-Me6). Others are sites from the La Thuile release not included in Fluxnet2015, which raises the possibility of data quality concerns (BW-Ma1, ES-ES1, RU-Zot, US-Bkg, US-SP3). But for the majority there is no immediately obvious explanation (AR-SLu, AU-Cpr, AU-Cum, AU-Gin, AU-Otw, AU-Tum, CN-Cha, CN-Cng, CN-Du2, DE-Seh, FR-Fon, US-AR1, US-AR2, US-Me2, ZM-Mon). While the data used in Fig. 11 is filtered for gap-filled and other quality control flags, we can confirm that using the entire time series for each site does not result in any qualitative change to site locations on this figure (see Fig. S2c).




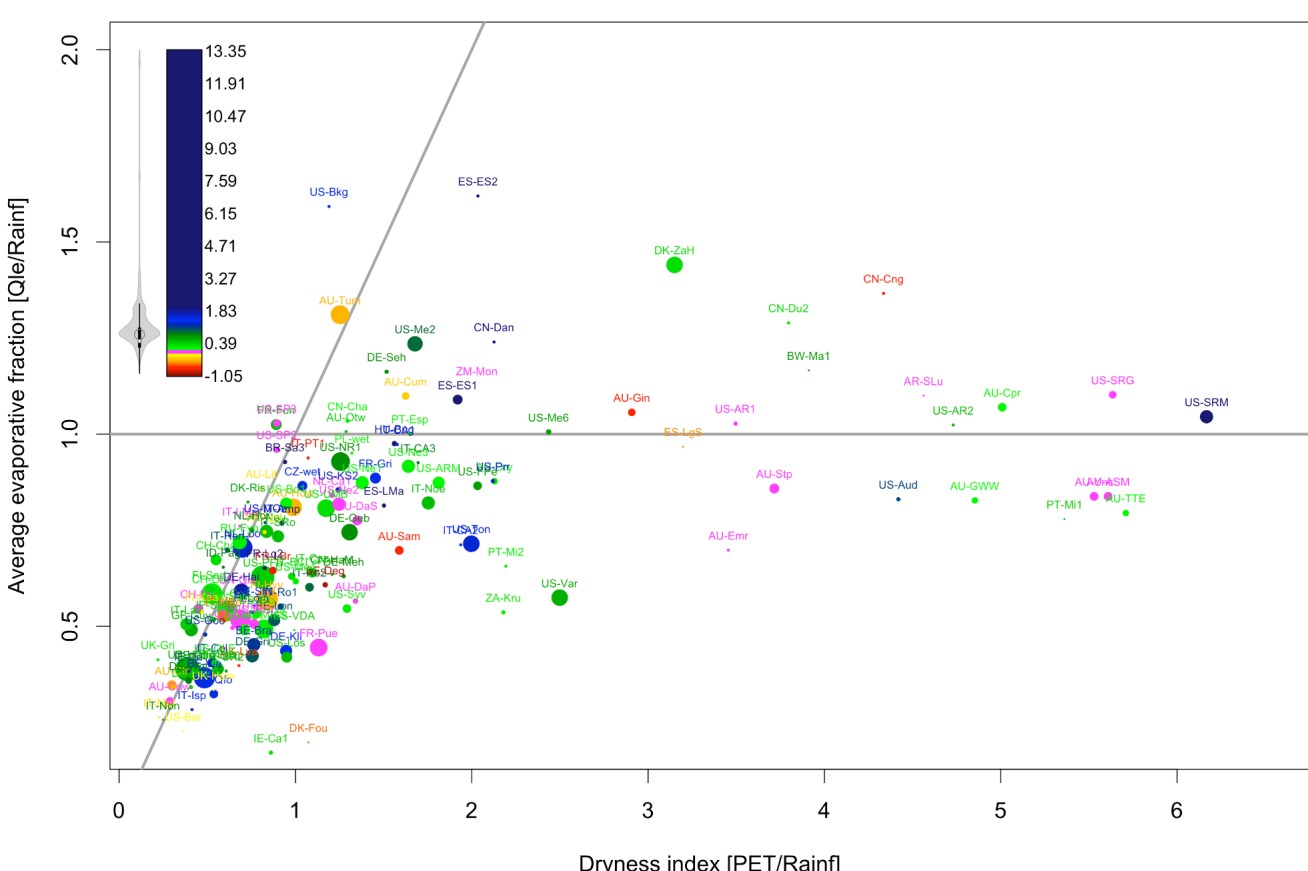

**Figure 11: Independent normalised metric value (iNMV) improvement offered by LSTM over the median iNMV value of all LMs (excluding empirical and physical benchmarks), shown by colour for latent heat flux (Qle). Each site's location is shown on axes of observed evaporative fraction versus dryness index. The prevalence of particular colour values is shown by the violin plot to the left of the colour legend. Values within [-0.1,0.1] are shown in pink, and values above 2 have constant, dark blue colour. Dot sizes indicate the length of site data, ranging from 1 (smallest) to 21 years (largest) - see Table S2 for site details.**

The equivalent plots to Fig. 11 for corrected-Qle, Qh, corrected-Qh and NEE are shown in Figures S14a-d, respectively. None of these appear to support the community's heuristic expectation that LMs' performance decreases with dryness. While there is a cluster of energy-limited sites where LMs consistently outperform LSTM_raw (red-orange-yellow dots), there are also several water-limited sites where LMs do well, and the worst simulated sites by LMs, shown in blue, seem evenly spread throughout the figures.



While it's clear that LSTM_raw broadly outperforms LMs at most sites, there are clearly some sites (red-orange-yellow) where
LMs on aggregate outperform the LSTM. This does not appear to be the case consistently across all three fluxes for any
particular site, however, or indeed any clear signal about the type of sites (in terms of vegetation type, dryness or available
energy) that are better simulated. This probably suggests that these outcomes may be more stochastic than the result of any
structural advantage the LMs might have.

Versions of this plot for each individual submitted model (and indeed any future submission) are available on
https://modelevaluation.org in the PLUMBER2 workspace.

**4 Discussion and conclusions**

It should be clear from the results above that the metric choices we make when evaluating LMs are critical. It might appear
from Fig. 3 that many LMs (CABLE, CHTESSEL, CLM, JULES, MATSIRO, MuSICA, ORCHIDEE, NoahMP) perform
better than the 3km27 model here for Qle, something that could represent progress since the original PLUMBER experiment
(where no models outperformed the 3km27 model for standard metrics - see Best et al, 2015). There are however some
differences here that mean PLUMBER and PLUMBER2 results are not directly comparable. First, the single set of metrics
we're using here is a combination of the 'standard', 'distribution' and 'extremes' based metrics used in PLUMBER, and the
infamous poor LM performance in PLUMBER was for the standard metrics set alone. Next, Fig. 3 uses (dependent) normalised
metric range, rather than ranks. We also have fewer models, and different models, in each panel that is used to calculate the
metric range, and results are calculated over 154 instead of 20 sites. It nevertheless remains true that Qh is much more poorly
predicted than Qle.

While of a similar performance standard to Qle prediction overall, NEE was notably underpredicted by LMs in a way that Qle
was not. While it seems obvious that a lack of site history in LM setup (noting that this information was not available) is the
cause for this, it's intriguing to see that empirical models (also not given this information) were able to predict NEE without
this bias, in most cases without any LAI information at all (Fig. 2). These empirical models were out-of-sample (they did not
use any data from the sites they predicted in their training). This is a categorical indication that importance of site history and
leaf area is significantly overstated in our LMs, and not nearly as important as we believe it is for flux prediction.

The importance of metric choices becomes clear when we consider the difference in apparent LM performance between dNMV
and iNMV. By excluding the LM we're evaluating from the criteria that define good or bad performance (the set of the three
empirical models) we define benchmark levels of performance that are independent of the land model being evaluated. It
means that when the LM is *much* better, or *much* worse than a priori expectations, it will get a score that is proportionally
much better or much worse. Using metric ranks or dNMV instead limited the cost of poor performance in the cumulative



metrics shown in PLUMBER style plots, and so gave an artificially positive indication of LM performance relative to the reference benchmark models.

While this might be confronting, it provides the community with a framework with which to assess the significance of proposed
improvements to LM performance, in a way that is relatively insensitive to metric choice, and critically, is based on demonstrated capacity for improvement. That is, when a LM is worse than an out-of-sample empirical model given the same predictors, we *know* that there is enough information provided to the LM to do better. We suggest that the summative analysis we present here using iNMV is a fairer, more comprehensive representation of LM performance than either the original PLUMBER paper or the dNMV versions of the same analyses.


It's also clear that, as with most model comparisons, the summary statistics presented in this paper do not give us any categorical indications about how to start improving models. They nevertheless allow, perhaps for the first time, to fairly account for some of the inevitable difficulties and eccentricities associated with using observed data. By evaluating performance relative to out-of-sample empirical estimates we can actually begin to quantify expectations of achievable LM
improvement, and pinpoint the circumstances in which this potential for improvement is most apparent. Given the number of LMs involved we did not actively explore these circumstances in detail here, since they are particular to each LM, but have nevertheless provided an approach to achieve this. Some clear indications are already evident from the sites shown in green and particularly blue in Figures 11, S13a,b,c,d. These are sites where we *know* that LM model prediction can be substantially improved, since an out-of-sample empirical model offers substantial performance improvements using the same predictors as
the LMs. These are of course the *average* discrepancy across all LMs, so the capacity for improvement at a particular site is likely to vary for different models. Equivalent figures for each individual model and variable can be found on modelevaluation.org in the PLUMBER2 workspace via the profile page for each submitted model output. Data and analysis code from this experiment are also available and we openly invite further analyses and contributions from the community.

Beyond a lower bound estimate of potential improvement, the hierarchy of empirical models we examined can also provide more nuanced information about performance expectations. The difference in performance between 6km729 and 6km729lag, for example, gives us an idea about the improvements in flux simulation we should expect from adding in model states such as soil moisture and temperature, rather than simply having an instantaneous response to meteorology (see Figures 8, 9 and S12a,b,c). The same is also true of the RF and LSTM, although they had slightly different predictor sets and architectures. The
simplest model, 1lin, also makes it clear that much of what we might heuristically regard as high model fidelity is a simple linear response to shortwave forcing (Figures S4, S5, S6 and perhaps most importantly Fig. 4). It should be abundantly clear that simple diagnostics can be very misleading and that defining 'good' model performance is inherently complicated. Without the empirical model hierarchy we've detailed here, judgements about LM performance would almost certainly be susceptible to confirmation bias.



## 4.1 Data limitations


The question of whether the Fluxnet2015 energy balance correction process is appropriate is not clear. Figure 1 and its equivalents in the supplementary material do seem to suggest that available energy in LMs is indeed higher than in raw observations (although a priori this is not evidence that the observations are wrong), but the energy balance corrected versions of this plot show an even larger discrepancy. Similarly, the differences between corrected and uncorrected water evaporative fraction (Figures S10a and S10b) show that corrected Qle fluxes look markedly different to almost all models. The plots based


on iNMV do seem to show that the correction process helps improve performance for several LMs. Nevertheless there is also more subtle evidence in the performance of the empirical models that gives us other, contradictory information. LSTM_raw is the best performing reference model in Fig. 4, and as expected, LSTM_eb, trained to match qualitatively different (energy-balance corrected) target data, does not perform as well against raw flux data as LSTM_raw. This is what we'd expect.


However, when we look at the reverse situation, using LSTM_eb as the reference model, and energy-balance corrected fluxes as the target data (shown in Fig. S8c), the situation is quite different. LSTM_raw performs worse for Qh, as expected, but it performs better than LSTM_eb for Qle. This tells us that unlike for Qh, a sophisticated ML model trained on the corrected Qle flux has no advantage predicting corrected Qle than the same ML model trained on raw fluxes - in fact it has a *disadvantage*. A similar result can be seen for 6km729lag. It is less sophisticated than LSTM_eb, and trained to predict the raw fluxes, yet it


outperforms LSTM_eb. This suggests that the correction to Qle makes these fluxes *less* predictable. It is likely categorically incorrect, whereas the correction to Qh may well add some value. This may suggest that the missing energy in uncorrected fluxes might be more likely to be in Qh fluxes (agreeing with other proposed correction approaches - see Charuchittipan et al., 2014).


The analyses also highlight the inherent complexity of real world simulation. We see a significant number of sites with an evaporative fraction greater than 1, which, despite being entirely physically plausible, is simply not possible for current LM process representations to replicate. It tells us that either (a) access to groundwater beyond gravity drainage is common, (b) below surface bedrock structure has a significant local hydrological effect, and/or (c) horizontal advection of moisture in soil (and locally on the surface) plays a significant role in moisture availability at the ~1km$^2$ spatial scale (i.e. flux tower fetch).


Very few global coupled models include any of these effects. It is very likely that almost all sites and indeed much larger spatial scales are affected by this same issue to varying degrees, even if their evaporative fractions do not appear to be anomalously high. This is supported by the remarkable result that LM performance is apparently not any worse for sites where evapotranspiration exceeds precipitation (Figures 11, S14a-d).

### Model complexity and under-constraint


As noted in the introduction, these LMs would likely produce better simulations if they were allowed to calibrate their parameters to each site's flux data, although how large that benefit would be, relative to allowing the empirical models to also use testing site data, is not at all clear. Our interest here, however, is the fidelity these models provide in global simulations, where they are typically given only coarse vegetation and/or soil type maps from which to extract all of their parameter values



for each grid cell. This is what both LMs and empirical models were given to make out-of-sample predictions. Previous work has also suggested that providing local information for calibration might result in much larger benefits for empirical models than LMs (see Abramowitz et al., 2008).

This raises the question of whether LMs are too complex for the level of fidelity they provide. It's at least theoretically possible, for example, that an LM is perfect, but because we are unable to precisely prescribe its parameters for these site simulations (and global simulations) we are actively hindering its ability to get the right result. What the out-of-sample empirical models show is that the information available in LMs' meteorological variables *alone* - without any description of what type of vegetation or soil might be at a given site, or indeed the reference height of the measurements - is enough to outperform all of the LMs. This is not to say that LMs could not perform better with more detailed site-specific information of course, but the way that they were run here was designed to mimic their application at global scales, and for that job they are considerably more complicated than is justified by their performance. A more detailed examination of how well LMs perform when given detailed site information would not simply require showing that metric scores for LMs improved when given this information, it would require that LMs come closer to outperforming ML approaches also provided with similar site-specific information.

There are of course other reasons why we might want complexity in a LM beyond improved performance, like the ability to infer the impacts of particular decisions on a broader range of processes within the land system. But it's nevertheless important to know the degree of predictability that's possible with the increasing amount of information that our models are provided with - what we're missing out on that is categorically achievable. The fact that it's been found that increasing model complexity shows little relationship to performance in some circumstances, even when additional site information is provided, should be concerning (Lipson et al, 2023). We also need to recognise that the many increases in sophistication that we might want to include to improve the representativeness of LMs (see Fisher and Koven, 2020) may come at a significant cost. The more degrees of freedom we have in a model, the more and broader range of observational data we need to effectively constrain it, the less able we are to pinpoint model shortcomings, and the more susceptible we become to getting the right answer for the wrong reasons (see Lenhard and Winsberg, 2010). A very crude statistical analogy might be that if we have a model with one process that is right 90% of the time, the model is 90% accurate. But if we have a model with 10 serial processes that are right 90% of the time, the model is $0.9^{10} = 35\%$ accurate.

## 4.2 Empirically-based land models

So there is a balance between complexity that includes the processes that we know are important, and the lack of observational/process constraint that we can provide when trying to simulate these at the global scale. A reasonable but perhaps challenging next question is whether we might build entirely empirically-based LMs. In some sense, the distinction between "physically" based and "empirical" models is artificial, as in this context our LMs contain many empirical approximations already (something that makes LMs somewhat different from their atmospheric and oceanic counterparts). Perhaps the best



approach is to start with the constraining data that is available at the global scale, and then build a model that is dependent

only upon what is available. That way the model will not be under-constrained in the way that current LMs are. We can already see from the results above that the fidelity of flux predictions would certainly rival existing LMs. But replacing a LM would require implementation of conservation equations and stores of heat and moisture, and while progress is being made towards this kind of endeavour (Kashinath et al., 2021; Shen et al., 2023), an operational implementation is not yet available.

Perhaps the biggest question to address, should this approach eventuate, is the ability of an empirically-based LM to perform

reliably under future climate change conditions. Some of the empirical models here might not be as appropriate as others for this purpose (those given remotely sensed LAI as an input for example, although a significant number of LMs are in this category too). If empirical models perform significantly better in today's climate, as we've seen above, then to nullify that performance advantage, the nature of the future change needs to (a) be outside the range of behaviour seen in the existing global training data envelope (here, be behaviour not observed at *any* flux towers historically), and (b) be different enough to

the existing training envelope to cancel the existing performance discrepancy. Pitman & Abramowitz (2005) provide some evidence that the magnitude of change might not be enough to bridge this gap. It seems reasonable to assume that the closer the spatial and temporal scale of the empirical model implementation is to the processes that determine variability, the closer the empirical model comes to actually approximating the processes involved, and quite plausibly, the more likely it is to be able to perform out-of-sample in future climates (assuming a broad enough training set). Given that flux tower data is at a

similar temporal and spatial scale to processes represented in LMs this seems plausible.

### 4.3 Next steps

The next steps for the community towards building LMs that better utilise the information available to them seem reasonably clear. Understanding the shortcomings of an LM is not a simple process, so moving away from in-house, ad-hoc model

evaluation towards more comprehensive, community built evaluation tools, where the efforts of those invested in model evaluation are available to everyone will be key. This will allow results to be comparable across institutions and routine automated testing to become part of the model development cycle. This will need to cover both global scales (e.g. ILAMB; Hoffman et al., 2016; Collier et al, 2018) and site-based process evaluation (e.g. modelevaluation.org; Abramowitz, 2012). In both cases, inclusion of empirical performance estimates, such as those shown here, will be key to distinguishing incremental

improvements from qualitative improvements in LM performance.

We are also building the suite of analyses shown in this paper into a model development testing pipeline available via https://modelevaluation.org. This will allow the community to replicate these exact analyses in routine automated testing as a part of repository check-ins. For more information about how to engage in this process, adopt and adapt these to your needs,

please contact the lead author. We also note that there were additional LM participants in this work that removed their submissions once the performance of their models relative to others was shared in draft stages of manuscript preparation.



Finally, there is obviously much, much more to explore in the PLUMBER2 dataset. Most participants submitted many more variables than were examined in this paper (and several came close to the list in Table S2). This paper nevertheless provides a
broad overview of the experimental design and preliminary results for key atmospheric fluxes. The vast majority of submissions to PLUMBER2, as well the forcing and evaluation data, are publicly available on https://modelevaluation.org as a community resource for further analyses, and we actively invite further collaborations to utilise the data set that this experiment has produced.

**Code and data availability**

Flux tower data used here are available at http://dx.doi.org/10.25914/5fdb0902607e1 as per Ukkola et al. (2022), and use data acquired and shared by the FLUXNET community, including these networks: AmeriFlux, AfriFlux, AsiaFlux, CarboAfrica, CarboEuropeIP, CarboItaly, CarboMont, ChinaFlux, Fluxnet-Canada, GreenGrass, ICOS, KoFlux, LBA, NECC, OzFlux-TERN, TCOS-Siberia and USCCC. The ERA- Interim reanalysis data are provided by ECMWF and processed by LSCE. The
FLUXNET eddy covariance data processing and harmonisation was carried out by the European Fluxes Database Cluster, AmeriFlux Management Project and Fluxdata project of FLUXNET, with the support of CDIAC and ICOS Ecosystem Thematic Center, and the OzFlux, ChinaFlux and AsiaFlux offices. All land model simulations in this experiment are hosted in modelevaluation.org, and to the extent that participants had no legal barriers to sharing these, are available after registering with modelevaluation.org. The analyses shown here were also performed on modelevaluation.org, using the codebase publicly
available at https://gitlab.com/modelevaluation/me.org-r-library.

**Author contribution:** Experimental design was conceived by GA with assistance from ML, MDK, AU, MC and wider community feedback. Data processing and analysis was developed and completed by GA, with input from ML, MDK, AU, and JCP. Empirical model simulations were completed by JCP, SH, GA, CB, JF and GN. Physical benchmark models were
build and run by MB and HR. Mechanistic land model simulations were completed by MDK, AU, JK, XL, DF, SB, GB, KO, DL, XW-F, CO, PP, NV, HR, MB, SM, TN, HK, YZ, YW, BS, YK, KC, AW, PA, MC, JO, SC, SZ and CF. The manuscript was written by GA, with feedback and iterations between all coauthors.

**Competing interests:** The authors declare that they have no conflict of interest.

**Acknowledgements**

G.A., S.H., J.C.P., A.U. and M. dK. acknowledge the support of the Australian Research Council Centre of Excellence for Climate Extremes (CE170100023). A.U. acknowledges support from the ARC Discovery Early Career Researcher Award (DE200100086). This research was undertaken with the assistance of resources and services from the National Computational
Infrastructure (NCI), which is supported by the Australian Government. B.S., Y.W. and Y.Z. acknowledge the support of the



Netherlands Organisation for Scientific Research (NWO) (WUNDER project, grant no. KICH1. LWV02.20.004), and the Netherlands eScience Center (EcoExtreML project, grant ID. 27020G07). JF acknowledges the support of NOAA Cooperative Agreement, Grant/Award Number: NA19NES4320002. Contributions by K.O. and D.L. are supported by the National Center for Atmospheric Research (NCAR), sponsored by the National Science Foundation (NSF) under Cooperative Agreement No.

1852977. Computing and data storage resources for CLM5, including the Cheyenne supercomputer (doi:10.5065/D6RX99HX), were provided by the Computational and Information Systems Laboratory (CISL) at NCAR. LSTM models were run with compute resources provided by NASA Terrestrial Hydrology Program, Grant/Award Number: 80NSSC18K0982. Noah-MP simulations were funded by NASA grant 80NSSC21K1731.

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
