# Peer review of "On the predictability of turbulent fluxes from land: PLUMBER2 MIP experimental description and preliminary results"

_EGUsphere, 2023_

## Author Response (AR1)

**Response to Reviewer 1**

*The manuscript by Abramowitz et al. examines the predictability of energy and carbon fluxes from land by using several empirical models as benchmarks to evaluate land models. Basically, the manuscript is an extended study of Best et al, 2015 (JHM) and Haughton et al, 2018 (GMD). The idea is of course very compelling and challenging. However, I have concerns about the current format of the manuscript.*

We appreciate the time and effort put into reviewing the paper, and are glad the reviewer found the idea compelling and challenging. We agree with many of the points raised below regarding formatting, and address each in turn.

*1) The manuscript focuses too much on describing the PLUMBER2 MIP experiment, but lacks literature reviews of advances in physical understanding of land flux controls, e.g., stomatal conductance controlling latent heat and processes controlling GPP and Res, both of which fundamentally determine NEE.*

Yes, we could of course add a discussion or literature review of the physical controls on land fluxes, and would be very happy to add that if the editor also agrees this is needed (noting however that it will add to the length of the manuscript). The reason it is not there is because the manuscript is fundamentally not focused on this question - it is about understanding the best approach to evaluate land models using tower data, which methodological choices when doing this could lead to qualitatively different results, and then ultimately how those choices should be made. These choices include metrics, how to create a useful summative indicator, establish model performance expectations *before* seeing simulations, whether fluxes should be energy balance corrected or not, and other aspects of data quality control. This is as opposed to using this MIP to actually better understand the natural system (also a laudable goal of course, and something that is being investigated in separate pieces of work using this data by others). That this focus was not apparent is a failure on our part. It is now much more clearly articulated - we address this issue in more detail in our response to (2) below.

The expectation that the aim of this paper is to further our understanding of the physical system is entirely reasonable for *Biogeosciences*. But we also feel that the focus on how methodological choices in model evaluation - that often appear trivially unimportant and go unquestioned - qualitatively change the nature of scientific inference, is particularly important for the *Biogeosciences* audience, who use land models for scientific inference regularly.

As suggested, we have shortened the methodology section, and indeed the results section as well. However, one of the main roles of this manuscript, as detailed in the title, is to describe and justify the experimental setup, so a considerable amount of detail is still dedicated to this, as it does not exist elsewhere. If the reviewer or editor feels that some of this information is superfluous we would indeed be willing to cut it from the manuscript, but to us at least, it's not clear *a priori* what should be cut, and we note that no specific suggestions were made. We want to reinforce that the level of detail reflects the reality that a great deal of thought on the part of many people was put into the construction of the many different aspects of this experiment that could qualitatively change the nature of the conclusions. We believe the

relevance and importance of this experimental detail are now apparent, since the paper is better framed, poses a direct aim, and is restructured to make apparent how we have addressed this aim.

*2) Although the manuscript is quite long (47 unedited pages), no concise scientific or research question is formulated in either the abstract or the introduction.*

We really do agree with this, and believe it is the most important concern raised by both reviewers. We also feel that with the focus of the paper now better articulated, the relevance and importance of its considerable methodological detail has become clearer. We have now, in the second paragraph of the paper, clearly stated our main aims:

*This paper focuses on a relatively simple question: how should we fairly assess the fidelity of land models? We aim to develop an evaluation framework that gives us confidence that LM evaluation is not partial - not dependent upon a particular metric, observational data choice, over-calibration or overfitting, a particular location or time, or subset of processes - that it is the closest we can reasonably expect to a summative understanding of the shortcomings or strengths of a particular model. This aim is the basis of a LM comparison experiment, PLUMBER2, and we use results from PLUMBER2 to illustrate the framework. It follows from the first Protocol for the Analysis of Land Surface Models (PALS) Land Surface Model Benchmarking Evaluation Project (PLUMBER; Best et al., 2015; Haughton et al., 2016), and addresses many of the shortcomings in its first iteration.*

*Our question is unapologetically methodological, since the consequence of getting the answer wrong is very real – we rely on LMs for a great deal of scientific inference and societally relevant predictions. We consider our aim in two parts. First, what kind of simulation environment allows for the best observational constraint of LMs, so that poor model performance might fairly be attributed to a LM? Second, how do we best structure an evaluation framework to give us confidence in this kind of attribution? We discuss these two questions in turn and highlight how the experimental framework of PLUMBER2 addresses them in a way that the original PLUMBER experiment could not.*

*[2a] I strongly recommend that the authors reduce the length of the manuscript to at least half of its current length.*

We feel that with the paper length actually halved we could do little more than describe the experimental setup. However, we have indeed shortened the methodology section, and reduced the number of results shown in the main manuscript. Figures 5, 8 and 9 from the original manuscript have been removed, and remaining results have been discussed in more detail. Reviewer 2 correctly noted that the discussion of some of the results was far too brief, and we feel that (a) better framing, motivation and articulation of the key foci of the paper (b) more detailed discussion of fewer results, and (c) better tying of these results to the conclusions drawn about these foci have made the shorter manuscript more cohesive in a way that its (reduced) length is justified.

*[2b] This study is actually an extended study of Best et al, 2015 (JHM) and Haughton et al. 2018 (GMD). Although the manuscript describes some key differences from previous publications, these differences and improvements do not convince readers the novelty of the current research.*

This comment, and the one in (6) below where the reviewer states that "*most of the information is repeated in previously published work*" clearly speaks to the need for us to better articulate the focus of the paper, as this is simply not true. We do accept that we need to better "convince readers the novelty of the current research".

There are many novel aspects to this work relative to the original PLUMBER experiment that make this work categorically different, and a powerful resource for the community going forward:

- It contains a broad hierarchy of machine learning-based benchmarks that quantify information available to land models about flux prediction, from linear regression to random forest to Long-Short Term Memory models that have their own internal states. This allows us to define benchmark levels of performance that are much stricter than in previous studies and can be reasonably interpreted as a lower-bound estimate of site predictability, individually tailored to each site.
- It includes a much broader range of ecosystems and climate zones, using 170 instead of 20 flux tower sites
- It addresses energy conservation issues in the flux tower data, and can actually draw clear conclusions about the validity of the correction approach used in Fluxnet2015
- It includes an independent suite of metrics, so metrics like mean bias,correlation and standard deviation aren't double counted (e.g. if RMSE were included)
- Critically, it includes significant work on a summative metric that is independent of the model being benchmarked. This means that the way that *a priori* expectations of model performance are defined does not require reference to the model being evaluated. It results in categorically different results, as evidenced in the difference between Figures 3 and 4.
- It uses a much broader range of models, including ecologically focused models, which are used by many *Biogeosciences* readers.
- Instead of *only* looking at a summative metric, results are explored through the graphical lenses of:
    - Budyko framework, which revealed site behavioural characteristics at > 30% of sites that land models are structurally unable to replicate
    - Water evaporative fraction and energy evaporative fraction
    - Water use efficiency
    - Vegetation type
    - Site length

This is not a comprehensive list, and none of these were investigated in the original experiment. To address this communication failure we have highlighted these differences more explicitly and consistently in the revised manuscript, including a table contrasting the two experiments, and discussed their implications towards addressing our better articulated aims.

*3) The main findings of this "preliminary" research are that LMs perform better at estimating NEE and Qle than Qh, and that online LMs outperform offline LMs and empirical models remarkably outperform LMs (in the abstract). The results are quite surprising to me. I wonder where this result comes from.*

It is an excellent question, and it is something we hope that this paper motivates the community to explore. It is also not something we can answer here, since the answer is different for each of the participating models, and different in different meteorological conditions, moisture regimes and ecosystems. Reviewer 2 notes that the "*difficulty with multi-model, multi-site evaluation exercises is the large amount of potential material that the authors have to boil down and synthesise, and the difficulty of identifying specific conclusions*". This is precisely why this first paper coming from the PLUMBER2 MIP needs to detail the experimental setup and focus on high-level results. It is providing a far more comprehensive and methodologically complete platform (relative to the original PLUMBER experiment) for the community to investigate why empirical models outperform mechanistic models by such a wide margin. Different groups are already preparing separate work that tries to get at specific aspects of this important question, like the wide range of responses to vapour pressure deficit across the model ensemble (in preparation).

Indeed by including a much clearer motivation of the methodological decisions made, testing a wide range of different assumptions more explicitly, and making the testing platform public, PLUMBER2 facilitates these kinds of critical analyses in a way that the first PLUMBER experiment could not. It is actively designed for this purpose.

An aside: noting the reviewers comment that "*online LMs outperform offline LMs*" - we want to be clear that all LM simulations were offline. It is true that those LMs built to be used in a coupled environment performed better, presumably what is being referred to here.

*4) The abstract should have a clear scientific question.*

Yes, as noted above, we do agree with this. In our case the question is quite fundamentally methodological - about the steps we need to take to ensure model evaluation is a true reflection of the predictive ability of a model (rather than data quality, overfitting, or metric choice). This might well be different to an expectation that we focus on what controls surface fluxes, but it is an important scientific question nonetheless. As noted in our response to [2] above where we have explicitly detailed new aims, discussed their motivations, and restructured the discussion and conclusions to address these.

*5) In the introduction section, most of the references cited are from the authors themselves and their own research groups. I believe that there is a lot of literature that is closely related to this research but is overlooked. Again, the scientific question and the aim of the research should be described in the introduction.*

As noted above in our response to [2] and [4], we agree with this and have addressed this in an entirely rewritten paper introduction, with more detailed referencing.

*6) The methodology section can be largely reduced as most of the information is repeated in previously published work. Please consider moving it (including Table 1) to the Supplementary Information section.*

This is perhaps where a lot of the misunderstanding comes from - we really do not agree that "*most of the information is repeated in previously published work*". Both PLUMBER and PLUMBER2 compare mechanistic models with empirical models at flux tower sites, that much is the same, we agree. So perhaps to someone working in a different area, or with different ideas about what mechanistic models represent, they may seem identical. But since the original PLUMBER paper in 2015, many arguments have been made dismissing the results as not representative of the state of mechanistic modelling. A great deal of work addressing these concerns and creating a robust platform to explore these results is embodied in this paper (including more than 8x the number of sites, machine learning benchmark hierarchy and more, as detailed in the list in our response to [2b] above).

The four sections of the methodology - flux tower data, land model simulations, machine learning benchmarks, and analyses - all detail new material that is not included in the papers the reviewer refers to, and do not repeat material from earlier papers.

Table 1 in the original manuscript details the models that participated in the experiment, as well as specific information about how each model conducted simulations. This is not described anywhere else, and the group of models is different (literally and qualitatively) to the previously published papers. We would suggest that if there is any interest in the physical understanding of land flux controls, as suggested above, this information is critical, at least in the modelling context.

To resolve this issue we have rewritten and/or edited all of these sections to more explicitly state what was done in the original PLUMBER experiment that the reviewer refers to, and highlight why the new approach that is taken in the paper resolves an important shortcoming in the original experiment. The model table and metrics table have now been moved to supplementary material.

*7) In the Results section, I note that all analyses are based on the average of all sites. We know that the partitioning of available energy differs largely across global flux sites. For example, a model error of 10 W/m2 at a wet site and a dry site means a big difference. Will treating the bias at all sites with equal weight affect the calculated metric value? Such difference may have impacts, particularly for PDF plot (Figure 5). In addition, some figures can be combined, e.g. Fig. 1-2 and Fig. 6-7. Such combinations make it easier for authors to follow the story.*

Only the analyses in Figure 1 and Figure 2 in the original manuscript - two out of eleven figures - average results across sites. Figure 6 through to Figure 11 looked at the spread of

results across different sites, looking at box plots for sites belonging to each vegetation type, evaporative fraction, water use efficiency, or each site's location on a Budyko curve, for example.

Nevertheless, the reviewer raises an excellent point regarding to the potential for unfair weighting between e.g. wet and dry sites, when an absolute error metric is used. This is precisely why in PLUMBER we have moved away from a traditional model intercomparison "evaluation" (comparing error metrics directly) and towards a "benchmarking" approach (comparing a model's relative performance to a benchmark that is specific to each site). Using benchmarking, the difficulty of prediction at "wet" and "dry" sites (or other differences, e.g. data quality) can be accounted for, and unfair weighting of sites avoided.

In Section 2.3 we state:

*"The empirical models we use as benchmarks are also listed in Table 1. As suggested above, these are key to quantifying site predictability, and so setting benchmark levels of performance for LMs that reflect the varying difficulty or complexity of prediction at different sites, unknown issues with data quality at some sites and more broadly understanding the amount of information that LM inputs provide about fluxes."*

However we agree that this key benefit of benchmarking could have be better introduced earlier in the manuscript, and believe we have rectified that in the rewritten introduction setcion.

Once again, this is a paper giving only high-level summative results of 170 separate simulations all completed by 32 different models of somewhere between two and fifty variables, after giving a complete experimental setup description. Examining results at each of the 170 individual sites, or coming to an understanding of a particular model's likely cause of discrepancies is simply not practical for this paper. More detailed, process-based, specific analyses in additional papers are indeed being prepared by others, as noted above, but we argue, do not fit in this context. Having this paper detail the motivation and methodology of the experiment, and how it addresses criticisms raised about the original PLUMBER experiment, is critical for future work to refer to and have the space remaining to explore process-level investigations in detail.

We have no *a priori* objection to combining the figures as suggested, but it would mean many full page figures. Reducing figure size would make it very hard to see the detail, and the required font sizes might make much of the material unreadable. We are happy to take advice from the editor on what the journal would prefer, but our preference would be to keep the figures at half page or less in size. Please also note that as we detail above, the number of figures has been reduced, perhaps resolving this concern anyway.

*8) In the Discussion section, I find that the discussion does not focus on the topic – whether the land fluxes can be predicted or not and their causes and how LMs can be improved to improve the flux prediction performance. I believe that the modelling community would welcome this type of discussion.*

Once again, we agree that this is an excellent topic of discussion, and that understanding the reason for this poor performance is very important. It is indeed the kind of discussion we hope that this experiment will engender in the community, and of course, we all want to know the answers to these questions. However, as noted above, the answers are not at all simple, specific to each model, and different in different conditions. The kind of analysis that this requires cannot be part of a paper that also details the considerable nuance in experimental setup that is required to ensure that results of a large model intercomparison like this are reasonable and fair. If this were a study involving one or two models, or just a handful of sites, we would agree that more detailed analysis is needed.

*9) Lines 230 and 645, it is stated that Qle/Rainfall is greater than 1 at more than 30% of the flux sites. This is quite plausible, as the Budyko framework was not used to assess the water balance at the site level, but at the watershed level, as explained in lines 646-647. However, I wonder whether the exclusion of these sites (Qle/Rainfall > 1) has a significant impact on the main results (Fig. 1-10).*

That information is readily available in Figure 11 already. If it did have a significant impact, there would be a marked difference in the colours above the (horizontal) evaporative fraction = 1 line: blue and green colours would be prevalent above, and red and yellow colours below. This is clearly not the case.

We have added the following text to the results section:

*None of these show a markedly higher density of poor LM performance (green-blue dots) above the 1.0 line where Qle exceeds precipitation on average. So despite there being a structural impediment to LMs simulating these sites, that impediment is clearly not the major cause of LM's poor performance*

as well as added to the discussion on this topic.

*10) If what I understand is correct, the "mean" flux is the sole focus covered in the current manuscript. Inter-annual variability and trend of land fluxes are also anticipated in the modelling community due to the large number of sites with long data (> 10 years).*

No, only the results in Figures 1, 2, 6 and 7 show results using mean fluxes. The others use the distribution of half hourly flux values in different ways, such as the density of half hourly values (Figure 5 in the original manuscript), and summative performance at the half hourly timescale in terms of temporal correlation (which includes interannual variability), performance in extremes (5th / 95th percentile difference), discrepancies in standard deviation, density estimate overlap and more.

Once again, we completely agree that inter-annual variability and trend would be interesting to examine. If this experiment looked at just a few models, or just a few sites, this kind of additional analysis could be feasible, but the interannual variability or trend in flux of 32 models at 100s of sites could only be cursorily summarised in a single figure. We also note that this is

a request to make the paper longer, at the same time as we are being asked to shorten the paper.

[10a] *Although the current manuscript has many significant weaknesses and some of which are due to the complexity of the problem under study, I will be happy to provide a further comprehensive assessment when the authors restructure the manuscript with far fewer pages, informative illustrations and focused scientific questions.*

While we disagree with several of the suggested changes, particularly the suggestion that the methodology is in essence superfluous, we nevertheless agree that the onus is on us to make this work more relevant to the community, so your further assessment would indeed be welcomed. We suspect the perspective you have provided is very much that of the broader *Biogeosciences* community, so that making sure we have communicated the important findings of this work in a clearer and more concise manner in our revised manuscript is in everyone's best interests.

*I hope that the authors do not take these comments as negative but rather as a way to improve the quality of the paper.*

Absolutely! We agree with many of these points raised, and have endeavoured to strengthen the paper accordingly.

**Response to Reviewer 2**

*This manuscript compares simulations of key fluxes from (mechanistic) land models with those from various empirical models (which importantly are not trained on the same data), using a large dataset of measured fluxes from sites around the world. Results include that the latent heat flux is generally better predicted than the sensible heat flux, with other results concerning the relative performance of different types of LM. There are also interesting indications that the energy closure approach used to adjust the measured fluxes might not be appropriate – though most results were insensitive to this aspect of the data. In general the empirical models outperform the mechanistic models, as in previous studies.*

*[1] This is a long paper (also with a substantial supplementary section) but despite this in some regards it is primarily setting out an approach that can be used to look at predictability, rather than posing and answering specific scientific questions in this area. (In this it is perhaps more a [long] technical note for Biogeosciences rather than a research article. Or would another journal be more appropriate?)*

We agree this is a reasonable question, and more or less agree with the assessment of the nature of this work. It would be within scope for *Geoscientific Model Development*, for example. Yet we feel that the central messages in this work (that we could have admittedly better communicated) are of critical importance to the *Biogeosciences* community as colleagues who most often utilise land model output in studies trying to further understanding of surface fluxes and associated processes.

*[2] A common difficulty with multi-model, multi-site evaluation exercises is the large amount of potential material that the authors have to boil down and synthesise, and the difficulty of identifying specific conclusions. The manuscripts authors acknowledge these issues, but I am not convinced that they have quite found a solution in this manuscript.*

Yes, we appreciate this acknowledgement, this was one of the most difficult aspects of writing this paper - the many possible dimensions to analyse, so that any particular figure, or indeed collection of figures for an entire paper, are necessarily only partial. Adding the hierarchy of machine learning approaches only made this harder. Coupled together with the community's familiarity or even expectation of detailed, process level analyses, it is quite a challenge! The additional perspectives of these two reviewers are clearly important for getting the balance right when refining the manuscript. We address each of the points raised below.

*[3] Although I am broadly familiar with the original PLUMBER exercise I do not know the details of this (and other work cited, such as that of Haughton) and this manuscript did not fill in those details for me. Rather the Introduction says that PLUMBER2 will be better and leaves it at that. The existing literature in general needs to be covered better – I don't want a long review, but I need more to provide motivation for PLUMBER2.*

Yes, this was also raised by Reviewer 1 and we do agree it is an issue. Both better contextualisation of this work and clearer communication of the central questions it poses to answer are needed. We have spent considerable effort on both of these tasks, in the form of revised text that directly compares PLUMBER and PLUMBER2. Below is an incomplete list of the novel aspects of this work relative to the original PLUMBER experiment that we have included in the revised manuscript in a table as well as in the body of the text:

- It contains a broad hierarchy of machine learning-based benchmarks that quantify information available to land models about flux prediction, from linear regression to random forest to Long-Short Term Memory models that have their own internal states. This allows us to define benchmark levels of performance that are much stricter than in previous studies and can be reasonably interpreted as a lower-bound estimate of site predictability, individually tailored to each site.
- It includes a much broader range of ecosystems and climate zones, using 170 instead of 20 flux tower sites
- It addresses energy conservation issues in the flux tower data, and can actually draw clear conclusions about the validity of the correction approach used in Fluxnet2015
- It includes an independent suite of metrics, so metrics like mean bias,correlation and standard deviation aren't double counted (e.g. if RMSE were included)
- Critically, it includes significant work on a summative metric that is independent of the model being benchmarked. This means that the way that *a priori* expectations of model performance are defined does not require reference to the model being evaluated. It results in categorically different results, as evidenced in the difference between Figures 3 and 4.
- It uses a much broader range of models, including ecologically focused models, which are used by many *Biogeosciences* readers.
- Instead of *only* looking at a summative metric, results are explored through the graphical lenses of:
  - Budyko framework, which revealed site behavioural characteristics at > 30% of sites that land models are structurally unable to replicate
  - Water evaporative fraction and energy evaporative fraction
  - Water use efficiency
  - Vegetation type
  - Site length

*[4] As I understand it, PLUMBER concluded that the flux of sensible heat was simulated better than that of latent heat, which was consistent with the idea that the latter was in some senses a simpler process. PLUMBER2 reverses that conclusion (albeit with different metrics), but the fact that this is essentially ignored in the manuscript is consistent with the scant coverage of PLUMBER in the text.*

No, in this sense the result is the same - sensible heat prediction is consistently worse in land models, despite being a conceptually simpler process in model representation. We have highlighted that this finding is reinforced in PLUMBER2, which explores a much more diverse range of environments (170 versus 20 sites, across more ecosystems) and more robust methodology (e.g. independent metric suite and summative benchmarking approach that is

independent of the model being benchmarked). This also raises the importance of highlighting the similarities and differences in the results of the two experiments, which we have now done in the revised manuscript.

[5] *My concerns around the amount of material and the low profile of specific questions were reinforced by the material on p26 around Fig.8. The paragraph describes the figure…but there is no mention of results. The next paragraph then moves on to related figures in the Supplement (this time with a little bit on results), but as a reader I was left wondering where any discussion of the implications of Fig.8 might be. Similarly on p30 there is discussion of Fig.11, including some interesting comments on the scale of applicability of the Budyko hypothesis, but it is almost an interesting aside and I was left looking for wider importance/more connection to the specific questions being studied.*

Yes, this is a reasonable criticism. We have dropped three of the figures (5, 8 and 9 from the original manuscript) and spent more time explaining the significance of those that remain. Our response to [6] below covers a similar issue.

[6] *Taken together Figs 3,4, 8, 9 and 10 contain many panels and a vast number of results, but it is arguable that little or no use is made of much of this information. The detailed results are most likely useful to the individual modelling groups, whereas the amount of information on display is almost overwhelming to general readers - and as noted much of it is not discussed in any detail. Ideally we might get a flavour of some of these results (e.g. for a subset of models?), but the full details could be left to the Supplement or modelevaluation.org.*

Yes, as noted above we have dropped Figures 5, 8 and 9 from the original manuscript, moved them supplementary material, and spent more time detailing the significance of those that remain. There is of course a balance to strike, since this work is indeed generating a lot of interest within the participant groups, but most model-specific results are now part of the supplementary material.

[7] *My overall feeling having read this manuscript was that there was too much material for the limited number of new results or conclusions. What was written was correct and generally presented well, but there was no strong sense of a message being conveyed. This volume of information might be appropriate under some circumstances, but at present it feels excessive in comparison to how little use is made of it - and the general impression is of this paper being a broad description of a possible framework or technique without specific questions or new conclusions. The text itself notes that the choice of metric can (in general) strongly influence conclusions – so we are left wondering what is really knew and conclusive in PLUMBER2..*

Yes, our feeling after reading these reviews is that this paper definitely needed a tighter focus - both in terms of explaining the context of the work, its aim, reducing the number of results presented in the main paper, and spending more time elucidating the nature and more importantly significance of what was found. While these were essentially text changes, they

are substantial and it took considerable time to reframe the paper appropriately. The introduction is entirely rewritten with clarified aims, methodology and results shortened, and discussion and conclusion refocused on the newly articulated aims.

[8] *I agree that there needs to be a place in the "literature" for papers that describe the development of both the models and the accompanying modelling methodologies (including benchmarking). And those papers can be difficult to write and find a home for. However, in this case I feel that a shorter, more focussed paper would be more useful for the wider audience, and that more of the more detailed results should be moved into the Supplement (or elsewhere).*

Yes, we agree, this is precisely what we have done, as detailed above.

*Minor comments*

[9] *The use of contracted forms such as "it's" and "we're" means the style is less formal than might be expected. I don't know if there are journal and/or editorial guidelines for this.*

We have reverted to a more formal tone throughout the manuscript.

[10] *P23 (Qle/Rainf) and p30 (iNMV etc.) – use brackets rather than dashes to delimit these, as the latter can be read as minus signs, which is confusing.*

No problem, we will amend instances of dashes where they might be confused for subtraction.

---

## Author Response (AR2)

**Response to Reviewer 3**

*Abramowitz and others critique methodologies used for model comparisons using outputs of the PLUMBER2 experiment. Honestly I like the conversational tone and general idea, but a few statements need more context and clarity. It's hearting to see that the modeling community is coming around to the idea that the best model for one site is just a simple model trained on a different site, and that the seemingly endless parameterization, boundless model complexity, and questionable spin up procedures is often overkill, at least when predicting fluxes over short time periods like half-hourly measurements to annual (or interannual) sums.*

The time taken to give your considered feedback is much appreciated - we clearly agree! We have endeavoured to respond to each point raised below. Some involve changes to the manuscript, and others are simply points of clarification.

*The abstract would be stronger if more quantitative. The first part of the pargraph beginning line 48 is an example (although the point in the latter part of the paragraph is extremely interesting).*

As suggested, we have amended the abstract to be more quantitative, starting with the paragraph starting on line 48:

> *"Predictions from 7 out-of-sample empirical models are used to quantify the information available to land models in their forcing data, and so the potential for land model performance improvement."*

> *"In all but two cases, latent heat flux and net ecosystem exchange of $CO_2$ are better predicted by land models than sensible heat flux, despite seeming to have fewer physical controlling processes."*

We appreciate these are only two minor instances (although they include the suggested change), but feel that including more detailed specific results here would detract from the ability to present the broader findings of this work.

*80: are these really shortcomings or is the best available approach at the time? As we learn more, many things become a shortcoming in retrospect but this might not be a fair comparison as innovation proceeds.*

Yes, no disagreement, but given this is more or less the same team of authors we're happy to wear the criticism!

*94: is the point here that pixel-scale estimates can never be validated?*

'Never' is perhaps a little strong, since this is clearly dependent on resolution, observational constraint, and the heterogeneity of the landscape within the grid cell. One could imagine a heavily instrumented 1km x 1km grid box that could be 'validated' (although I dislike this term, as it is categorical), or alternatively, gridded evaluation products that truly attempted to quantify observational uncertainty in their estimates (rather than, say, the spread of available products). Our point here is really just that at this point in time, model constraint is much closer to being achievable at the site scale.

*109: this is a defined problem in and of it is related to the subpixel heterogeneity problem*

Yes, heterogeneity will always be a challenge for any gridded simulation, as it means that model simulations must represent emergent characteristics rather than be an explicit representation of the surface. But this section is discussing whether it is reasonable to evaluate models that are *inherently designed for gridded scale simulation* at the higher resolutions of their intended applications.

*178: some of the sites may have time-evolving data but there isn't an easy structure in flux network data compilations that make it easy to include it*

Yes, understood. It's also not likely to be available at a high enough proportion of sites for this kind of broad-scale modelling experiment.

*195: why wind speed, was friction velocity or the standard deviation of vertical velocity (sigma_w) not considered for data quality control?*

Wind speed was used as a simple, easily available proxy. It does not need to be perfect, as it is just being used to understand whether or not the apparent poor performance of land models was in fact due to aggregate metrics being dominated by low turbulence periods… it did not make any qualitative difference.

*222: I'm happy to see that the authors critiqued energy balance closure-based "corrections" (quotes intentional) to data and not surprised to find that adjusting data lowered its quality. The flux and modeling community really needs to stop adjusting flux data because the lower-frequency flux transporting eddies responsible for much of the lack of closure needn't be strongly related to turbulent fluxes from the footprint.*

Yes, we agree 'corrections' should not be applied universally when the basis for these is not proven to be sound - it will only make prediction of 'observed' fluxes more convoluted and difficult. We do believe that some version of the machine learning approach here could form the basis of a metric that might go part way to discerning whether a given 'correction' is appropriate or not.

*449: 'there is no a priori reason'…yes there's a very strong a priori reason because both Qle and NEE are dominated by stomatal function (transpiration and GPP). There should be a strong relationship.*

Yes, sure, we have amended this text to read:

> *"Given the expectation that NEE is likely to be strongly dependent on site history, and that we could not reliably include this information in the modelling protocol or account for it in this plot, there is no a priori reason to expect a clear relationship across all sites here, beyond both fluxes being dependent on stomatal function."*

*455: regarding this, were sites always compared against NEE measurements that include subcanopy storage for forest systems, or were eddy flux ('Fc') measurements used as a surrogate for NEE? This is a great approximation for non-forested ecosystems but there will be a large bias, especially when partitioning GPP and RE, in many forests if storage flux isn't considered.*

Yes, eddy flux measurements were used, but we did not partition GPP and RE.

*Figure 5: Why wasn't the random forest model included? From personal experience, XGBoost often emerges as the best fit for flux data, which is most similar to RF.*

The random forest model is included. It is in the top row of subpanels, labelled "RF_raw" and "RF_eb". Perhaps the reviewer is asking why RF was not used as the highest performing reference model, since XGBoost worked well for them…? The answer, as noted in the text and shown in the figures, is that the LSTM outperformed the RF model out-of-sample, most likely because it includes internal states.

*Why is color faded in some Fig 6 subpanels?*

Figures 5 and 6 both have the first 10 subpanels faded. As noted in the caption "*The first 10 panels (faded) show empirical or physical benchmark models.*"

*55: I'm not entirely convinced that this is a lower bound estimate. For a real lower bound wouldn't one want a method rooted in information theory that can estimate a theoretical predictability? I know that this is more of a thing for meteorology and perhaps not pertinent to this discussion, but perhaps one day it might be (e.g. 'predictive power' from Schneider & Griffies 1999: https://journals.ametsoc.org/view/journals/clim/12/10/1520-0442_1999_012_3133_acffps_2.0.co_2 .xml)*

We assume this refers to the use of 'lower bound' on line 559. The reviewer is correct that we could work harder to produce a better empirical model that could predict fluxes with more fidelity, and so provide a stricter, higher estimate of the lower boundary of predictability - this will always be true. Whether considered in an information theory framework or not, the process in this case will always be empirical rather than analytical, in a mathematical sense. That does not change the fact that 'true' predictability is higher than our best empirical model here – it remains a lower bound estimate.

*Fig 8 I'd like to think that almost all wetland sites will have an evaporative fraction greater than 1; considering these as a unique case could provide even more context.*

Interesting point, we've now mentioned wetland sites in two places in this discussion. Actually none of the sites described as wetland have an evaporative fraction greater than 1. But we would only really expect this if they were in a water-limited climate. Cold area wetlands presumably do not have the available energy to evaporate more water than is precipitated. Text changes are:

> *"A significant proportion of sites had Qle fluxes larger than incident rainfall, and since this is something that most LMs will be structurally prohibited from replicating (with the possible exception of wetlands), we explore why this might be the case, and whether the issue has biased our overall conclusions about LM performance."*

> *"Of the sites in Fig. 8 with water evaporative fraction greater than 1, only one is irrigated (ES-ES2) and none are wetland sites."*

*676: I'm not against the statement that the correction is categorically incorrect, per se, but feel that the authors are stating that it's categorically incorrect for the incorrect reason. Between unmeasured storage flux terms including photosynthetic energy flux, mesoscale meteorological*

*motions, and the fact that advective flux does not necessarily follow the Bowen ratio correction, there are many physical, i.e. not modeling reasons for why the correction is incorrect.*

We do not provide any physical reasoning for why the correction is categorically incorrect, just point out that we have a metric that can show it is. That does not make our metric "the incorrect reason".

As an analogy, if a bath is overflowing because the plug is left in, it is not incorrect to state that we can measure the rate or amount of water flowing over the side. Many true statements can be made about this situation - for example that we can identify that there is a problem because the water is flowing out - that are not about identifying the cause.

*760: only if the errors are independent*

Yes, of course. It is, as we stated, a 'crude analogy'. We have now added to this:

> *"A very crude statistical analogy might be that if we have a model with one process that is right 90% of the time, the model is 90% accurate. But if we have a model with 10 serial processes that are right 90% of the time, the model is 0.910 = 35% accurate (although only if errors are independent)."*